# Subspace-Guided Continual Learning: Hessian Based Stable–Plastic Decomposition for Exemplar-Free Class-Incremental Learning

## Abstract

Exemplar-Free Class-Incremental Learning (EFCIL) presents a significant challenge in continual learning, where a model must learn new classes sequentially without access to old data, making it susceptible to catastrophic forgetting. The core difficulty lies in balancing model stability (preserving old knowledge) and plasticity (acquiring new knowledge). We propose Subspace-Guided Continual Learning (SGCL), a novel method that tackles this dilemma from a geometric perspective. SGCL functionally decomposes the feature space into two orthogonal subspaces: a "stable subspace" containing feature directions critical for previous tasks, and a "plastic subspace" where new knowledge can be learned with minimal interference. We demonstrate that this decomposition can be efficiently identified by analyzing the feature-space Hessian, where its high-curvature eigendirections define the stable subspace. Building on this, SGCL introduces two synergistic components: 1) Subspace-Guided Regularization (SGR), which imposes strong, curvature-weighted penalties on feature drifts within the stable subspace, and 2) Subspace-Guided Prototype Alignment (SGPA), which adaptively corrects the shift of old-class prototypes to recalibrate the classifier. Extensive experiments on standard benchmarks, including CIFAR-100, Tiny-ImageNet and ImageNet-Subset, show that SGCL significantly outperforms existing state-of-the-art methods. Our work provides a principled and effective approach to EFCIL, offering a new perspective on mitigating forgetting by analyzing the loss landscape structure.

## 1 Introduction

Continual learning (CL) addresses the challenge of learning from a continuous stream of data without suffering from catastrophic forgetting (McCloskey & Cohen, 1989; French, 1999)—the tendency of neural networks to abruptly lose knowledge of previously learned tasks. This requires a delicate balance between model stability (preserving old knowledge) and plasticity (acquiring new knowledge), a conflict often referred to as the stability-plasticity dilemma (Grossberg, 1982; Mermillod et al., 2013). Major CL strategies are broadly categorized into regularization-based, replay-based, and architecture-based methods (De Lange et al., 2021; Masana et al., 2024; Zhou et al., 2023). Regularization-based methods add constraints to the loss function to prevent drastic weight changes; replay-based methods store and re-train on a small subset of past data; and architecture-based methods dynamically modify or expand the network structure to accommodate new knowledge.

A challenging yet practical paradigm is Exemplar-Free Class-Incremental Learning (EFCIL). In this setting, a model must learn new classes sequentially without storing any data exemplars from past tasks. Furthermore, during inference, privileged information such as task identifiers is unavailable. This constraint is crucial for real-world applications with strict memory budgets or data privacy regulations (Rebuffi et al., 2017; Gomez-Villa et al., 2024). The absence of past exemplars significantly exacerbates catastrophic forgetting, making it a formidable research problem. We address the cold-start (Magistri et al., 2024) EFCIL scenario, where the model is trained from scratch on an initial set of classes and incrementally updated, with classes evenly distributed across tasks.

Early EFCIL methods focused on parameter space regularization. Kirkpatrick et al. (2017) computes Fisher Information Matrix to identify important parameters, while methods like Wang & Zhang

(2023) use gradient projection to constrain updates within orthogonal subspaces. However, the high dimensionality of parameter space (often millions of parameters) makes accurate importance estimation costly and prone to approximation errors. This has motivated recent advances toward feature space management, where the lower dimensionality enables more precise control. Methods directly operating in feature space (Magistri et al., 2024; Petit et al., 2023; Goswami et al., 2023; Rypeść et al., 2024) implicitly or explicitly preserve crucial feature dimensions for old tasks. Building on this insight, we propose a principled theoretical framework based on the geometric structure of the classification loss landscape. Our key insight is that the feature-space Hessian of the cross-entropy loss naturally reveals which feature directions are most critical for preserving learned decision boundaries. The eigenvectors with large eigenvalues indicate directions of high curvature where changes would most significantly affect classification performance on past tasks. Furthermore, for a $K$-class linear classifier, the cross-entropy Hessian has rank at most $K - 1$, providing both theoretical justification and computational efficiency for our subspace decomposition. This principled approach separates *stable* directions (high curvature, requiring preservation) from *plastic* directions (low curvature, allowing adaptation).

Building upon this geometric insight, we propose Subspace-guided Continual Learning (SGCL), a novel EFCIL method that operationalizes the Hessian-based analysis through explicit feature space decomposition. SGCL identifies and separates a *stable subspace*, containing feature directions with high loss curvature that are critical for preserving past knowledge, from its orthogonal *plastic subspace*, which encompasses directions with low curvature that can safely adapt to new information. This principled decomposition enables two synergistic components: a Subspace-Guided Regularization (SGR) loss that selectively penalizes feature drift only within the stable subspace with weights proportional to the corresponding Hessian eigenvalues, and a Subspace-Guided Prototype Alignment (SGPA) mechanism that leverages the same geometric principles to modulate prototype updates for precise drift correction. Together, these components enable SGCL to achieve a superior balance between stability and plasticity in the demanding cold-start EFCIL setting.

The main contributions of this work are threefold:

- A novel Subspace-Guided Regularization (SGR) strategy that orthogonally decomposes features into stable and plastic subspaces, applying selective regularization to precisely balance stability and plasticity.

- An efficient stable subspace identification algorithm that exploits the intrinsic low-rank structure of the feature-space Hessian, avoiding expensive matrix decomposition.

- A Subspace-Guided Prototype Alignment (SGPA) mechanism that modulates prototype updates based on stable subspace projections for accurate drift correction.

## 2 RELATED WORK

### 2.1 CLASS-INCREMENTAL LEARNING METHODS

Class-Incremental Learning (CIL) methods are designed to learn new classes over time. They are often grouped into three main families (De Lange et al., 2021; Masana et al., 2024; Zhou et al., 2024).

**Regularization-based** methods introduce additional loss terms to penalize changes to parameters or representations critical for past tasks. Seminal works like EWC (Kirkpatrick et al., 2017), SI (Zenke et al., 2017), and MAS (Aljundi et al., 2018) estimate parameter importance, while feature-level regularization, such as knowledge distillation (Hinton et al., 2015), has proven highly effective. Learning without Forgetting (LwF) (Hou et al., 2019; Douillard et al., 2020).

**Replay-based** methods store a small subset of past data (exemplars) in a memory buffer to rehearse when learning new tasks (Rebuffi et al., 2017; Castro et al., 2018; Belouadah & Popescu, 2019; Li et al., 2024). While highly effective, this approach is not always feasible due to memory or privacy constraints. To circumvent the need for storing real data, some methods employ generative models to create synthetic samples of past data (Shin et al., 2017; Smith et al., 2021).

**Architecture-based** methods dynamically adapt the model's architecture, for instance by freezing parts of the network and allocating new parameters for new tasks (Mallya & Lazebnik, 2018; Yoon et al., 2018; Rypeść et al., 2023).

## 2.2 EXEMPLAR-FREE CLASS-INCREMENTAL LEARNING METHODS

**Parameter-Space Regularization** Early EFCIL approaches focus on constraining parameter updates to preserve learned knowledge. EWC (Kirkpatrick et al., 2017) and its variants compute importance weights via Fisher Information Matrix, penalizing changes to critical parameters. Gradient projection methods (Saha et al., 2021; Wang et al., 2021; Wang & Zhang, 2023; Zhao et al., 2023) take a more restrictive approach, constraining gradient updates to orthogonal subspaces to avoid interference with past tasks. While providing strong theoretical guarantees, these methods suffer from computational complexity in high-dimensional parameter spaces and can be overly restrictive, limiting plasticity and hindering beneficial knowledge transfer (Chaudhry et al., 2020).

**Feature-Space Regularization** Operating in lower-dimensional feature space enables more precise control over knowledge preservation. Knowledge distillation methods (Hou et al., 2019; Douillard et al., 2020) preserve feature distributions by matching outputs between old and new models. Elastic Feature Consolidation (EFC) (Magistri et al., 2024) identifies important feature directions via an Empirical Feature Matrix (EFM) and applies anisotropic regularization. While effective, these approaches rely on empirical correlations rather than principled loss geometry. Our SGCL method addresses these limitations by explicitly managing feature drift through *feature-space Hessian* analysis, directly capturing the curvature structure of the loss landscape with theoretical guarantees.

**Prototype Drift Correction** As feature spaces evolve during continual learning, class prototypes drift from their original positions, causing severe misclassification (Zhu et al., 2021a). FeTrIL (Petit et al., 2023) freezes the feature extractor and translates old prototypes via geometric transformations from new class features. FeCAM (Goswami et al., 2023) employs Mahalanobis distance to account for class covariance structures. LDC (Gomez-Villa et al., 2024) learns explicit mappings between old and new feature spaces, while ADC (Goswami et al., 2024) generates pseudo-samples through adversarial attacks. However, these methods tend to treat prototype drift in isolation, overlooking its intrinsic coupling with feature drift. Our method adopts a unified framework to jointly manage both feature drift and prototype drift through Hessian-based subspace decomposition.

## 3 METHOD

In this section, we first present the necessary preliminaries, we then introduce our Subspace-guided method for EFCIL, including Subspace-Guided Regularization (SGR), an efficient stable subspace identification algorithm, and Subspace-Guided Prototype Alignment (SGPA).

### 3.1 PRELIMINARIES

**Problem Formulation** We consider the Exemplar-Free Class-Incremental Learning (EFCIL) setting, where a model sequentially learns from $T$ distinct tasks $\{\mathcal{T}_1, \mathcal{T}_2, \ldots, \mathcal{T}_T\}$. Each task $\mathcal{T}_t$ contains its own set of classes $\mathcal{C}_t$ and training data $\mathcal{D}_t$. The model consists of two components: a feature extractor (backbone) $f_\theta : \mathbb{R}^{d_{\text{in}}} \to \mathbb{R}^d$ parameterized by $\theta$, and a classifier head $W$ that expands with each new task. Specifically, at task $t$, $W_t \in \mathbb{R}^{c_t \times d}$ where $c_t = \sum_{k=1}^{t} |\mathcal{C}_k|$ denotes the total number of classes observed up to task $t$. The key challenge of EFCIL lies in its strict data access constraint: at time step $t$, the model can only access the current task's data $\mathcal{D}_t$, while all previous data $\{\mathcal{D}_k\}_{k=1}^{t-1}$ remains completely inaccessible (Zhu et al., 2021c; Mai et al., 2022). Despite this constraint, EFCIL methods aim to approximate the performance of an ideal model trained jointly on all data. This intractable objective serves as a performance upper bound and is formulated as:

$$(\theta_t^*, W_t^*) = \arg\min_{\theta_t, W_t} \sum_{k=1}^{t} \mathbb{E}_{(\mathbf{x},y) \sim \mathcal{D}_k} \left[ \mathcal{L}_{\text{ce}} \left( W_t f_{\theta_t}(\mathbf{x}), y \right) \right], \tag{1}$$

where $\mathcal{L}_{\text{ce}}$ denotes the cross-entropy loss, $\mathbf{x}$ is an input sample, and $y$ is its class label.

Figure 1: Overview of Subspace-guided Continual Learning (SGCL). (a) Hessian-based subspace decomposition identifies stable directions for preserving past knowledge and plastic directions for learning new tasks. (b) Subspace-guided regulation penalizes feature drift ($\Delta\mathbf{z}$) in stable directions, preventing the new model ($f_t$) from forgetting knowledge of the old model ($f_{t-1}$). (c) Classifier calibration updates the classifier ($W_t$) using aligned prototypes and features from current data ($\mathcal{D}_t$).

**Projection Decomposition of Inner Product Spaces** For any feature vector $\mathbf{z} \in \mathbb{R}^d$ and subspace $\mathcal{S} \subseteq \mathbb{R}^d$, the orthogonal decomposition yields:

$$\mathbf{z} = \mathbf{z}_{\mathcal{S}} + \mathbf{z}_{\mathcal{S}^\perp}, \tag{2}$$

where $\mathbf{z}_{\mathcal{S}} = \text{Proj}_{\mathcal{S}}(\mathbf{z})$ and $\mathbf{z}_{\mathcal{S}^\perp} = \mathbf{z} - \mathbf{z}_{\mathcal{S}}$. Given an unit orthonormal basis $\{\mathbf{u}_i\}_{i=1}^k$ for $\mathcal{S}$, the projection is computed as:

$$\text{Proj}_{\mathcal{S}}(\mathbf{z}) = \sum_{i=1}^{k}(\mathbf{z}^\top \mathbf{u}_i)\mathbf{u}_i. \tag{3}$$

This decomposition enables selective regularization on different subspaces.

### 3.2 Core Principle: Functional Decomposition of Feature Space

Our key assumption is that the feature space can be functionally decomposed into two orthogonal subspaces based on their importance for preserving past knowledge: $\mathbb{R}^d = \mathcal{S} \oplus \mathcal{P}$, where $\mathcal{S}$ is the *stable subspace* containing directions crucial for preserving past knowledge, and $\mathcal{P}$ is the *plastic subspace* providing degrees of freedom for new learning. For convenience, we consider the feature space as the entire $d$-dimensional space.

Based on this assumption, we identify these subspaces using the feature-space Hessian $\mathbf{H}_f = \nabla_{\mathbf{z}}^2 \mathcal{L}_{\text{ce}}$, which captures the loss curvature of old tasks in feature space. While the Hessian involves second-order derivatives, for cross-entropy loss with feature $\mathbf{z} = f_{\theta_{t-1}}(\mathbf{x})$ and classifier $W_{t-1}$, it can be analytically computed (see Appendix A for a detailed derivation):

$$\mathbf{H}_f = \mathbb{E}_{\mathbf{x} \sim \mathcal{D}_{t-1}} \left[ W_{t-1}^\top (\text{diag}(\mathbf{p}) - \mathbf{p}\mathbf{p}^\top)W_{t-1} \right], \tag{4}$$

where $\mathbf{p} = \text{softmax}(W_{t-1}\mathbf{z})$. Importantly, this Hessian matrix has low rank:

**Proposition 1** (Rank of Feature-Space Hessian). *For a c-class classification problem with $W_{t-1} \in \mathbb{R}^{c \times d}$, if $\mathbf{p}$ has strictly positive entries, then the rank of $\mathbf{H}_f$ satisfies:*

$$r := rank(\mathbf{H}_f) \leq \min(rank(W_{t-1}), c-1). \tag{5}$$

*Moreover, when $W_{t-1}$ has full row rank, we have $r = c - 1$.*

This provides a clear rationale for our method. Therefore, the stable subspace $\mathcal{S} = \text{Im}(\mathbf{H}_f)$ is spanned by eigenvectors with non-zero eigenvalues (high curvature directions), while the plastic subspace $\mathcal{P} = \text{Ker}(\mathbf{H}_f)$ corresponds to zero eigenvalues (low curvature directions). While this decomposition is intuitively motivated by curvature analysis, we provide a theoretical analysis in Appendix D demonstrating that this choice minimizes forgetting bounds under the SGR constraint and local quadratic approximation, among all eigen-aligned subspaces with the same dimension (Theorem 1 and 2).

### 3.3 EFFICIENT STABLE SUBSPACE IDENTIFICATION

Directly decomposing the $d \times d$ Hessian $\mathbf{H}_f$ is computationally prohibitive. We overcome this by leveraging the low-rank structure: since $\mathbf{H}_f = W_{t-1}^\top \mathbf{A} W_{t-1}$, the stable subspace $\mathcal{S}$ is contained within the row space of $W_{t-1}$.

**Proposition 2** (Efficient Subspace Computation). *Let the $QR-$decomposition of the transposed weight matrix be $W_{t-1}^\top = \mathbf{QR}$, where $\mathbf{Q} \in \mathbb{R}^{d \times c}$ is an orthonormal basis for the row space of $W_{t-1}$. The non-zero eigenvalues of $\mathbf{H}_f$ can be found by decomposing a much smaller reduced Hessian, $\mathbf{H}_{red} \in \mathbb{R}^{c \times c}$, defined as:*

$$\mathbf{H}_{red} = \mathbf{RAR}^\top, \tag{6}$$

*where $\mathbf{A} = \mathbb{E}_{x \sim \mathcal{D}_{t-1}}[diag(\mathbf{p}) - \mathbf{p}\mathbf{p}^\top]$.*

1. ***Equivalence:*** *The set of non-zero eigenvalues of $\mathbf{H}_f$ is identical to the set of non-zero eigenvalues of $\mathbf{H}_{red}$. If $(\sigma, \mathbf{u})$ is an eigen-pair of $\mathbf{H}_{red}$, then $(\sigma, \mathbf{Qu})$ is a corresponding eigen-pair of $\mathbf{H}_f$.*

2. ***Efficiency:*** *The computational complexity of finding $\mathcal{S}$ via $\mathbf{H}_{red}$ is $O(dc^2 + c^3)$. This avoids forming and decomposing the full $d \times d$ Hessian $\mathbf{H}_f$, an operation with complexity $O(d^2c + d^3)$, making our method substantially more efficient for $c \ll d$.*

This proposition enables efficient extraction of the stable subspace without expensive full eigen-decomposition. Let $\{(\sigma_i, \mathbf{u}_{r,i})\}_{i=1}^r$ be the non-zero eigen-pairs from $\mathbf{H}_{red}$, where $r$ is defined in Proposition 1. The stable subspace basis vectors are $\mathbf{u}_i = \mathbf{Q}\mathbf{u}_{r,i}$ with corresponding curvature weights $\sigma_i$.

As shown in Figure 2, $QR-$decomposition of $W_{t-1}^\top$ yields $\mathbf{Q}$, which transforms the full Hessian problem into a smaller $c \times c$ eigendecomposition, making the computation substantially more efficient.

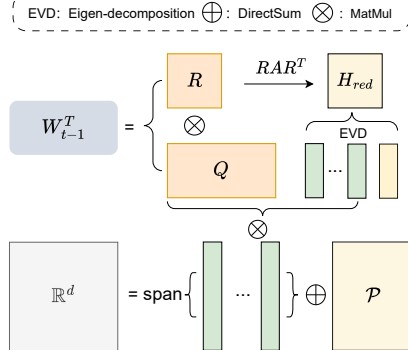

Figure 2: Efficient computation

### 3.4 SUBSPACE-GUIDED REGULARIZATION(SGR)

Once the stable subspace $\mathcal{S}$ is identified, we can apply differentiated regularization to feature drift $\Delta \mathbf{z} = f_\theta(\mathbf{x}) - f_{\theta_{t-1}}(\mathbf{x})$. Our strategy heavily penalizes drift within $\mathcal{S}$ while allowing flexibility in the plastic subspace $\mathcal{P}$.

**Stable Subspace Regularization.** We heavily penalize projections of the drift onto stable directions, weighted by their corresponding eigenvalues $\sigma_i$:

$$\mathcal{L}_{\text{stable}} = \mathbb{E}_{\mathbf{x} \sim \mathcal{D}_t}\left[\lambda_s \|\text{Proj}_{\mathcal{S}}(\Delta \mathbf{z})\|_{\mathbf{\Sigma}}^2\right] = \mathbb{E}_{\mathbf{x} \sim \mathcal{D}_t}\left[\lambda_s \sum_{i=1}^r \sigma_i \|\text{Proj}_{\mathbf{u}_i}(\Delta \mathbf{z})\|_2^2\right], \tag{7}$$

where $\| \cdot \|_{\mathbf{\Sigma}}^2$ denotes the weighted norm with eigenvalues $\sigma_i$.

**Plastic Subspace Regularization.** For the plastic subspace, we apply minimal uniform regularization:

$$\mathcal{L}_{\text{plastic}} = \mathbb{E}_{\mathbf{x} \sim \mathcal{D}_t}\left[\lambda_p \|\text{Proj}_{\mathcal{P}}(\Delta \mathbf{z})\|_2^2\right] = \mathbb{E}_{\mathbf{x} \sim \mathcal{D}_t}\left[\lambda_p \left\|\Delta \mathbf{z} - \sum_{i=1}^r \text{Proj}_{\mathbf{u}_i}(\Delta \mathbf{z})\right\|_2^2\right], \tag{8}$$

where $\lambda_p \ll \lambda_s$. The key difference is that stable regularization uses curvature-weighted penalties ($\sigma_i$) to preserve critical directions, while plastic regularization applies uniform minimal constraints. Then the total loss for feature adaptation combines the SGR penalties with the standard cross-entropy loss on the current task's data:

$$\mathcal{L}_{\text{adapt}} = \mathbb{E}_{(\mathbf{x}, y) \sim \mathcal{D}_t}\left[\mathcal{L}_{\text{ce}}(W_t f_\theta(\mathbf{x}), y)\right] + \mathcal{L}_{\text{stable}} + \mathcal{L}_{\text{plastic}}. \tag{9}$$

---

**Algorithm 1** Subspace-Guided Continual Learning (SGCL)

---

**Require:** Sequence of task data $\{\mathcal{D}_t\}_{t=1}^T$, initial model $f_{\theta_0}$ and classifier $W_0$.
1: **// Task 1: Initial Training**
2: Update $\theta_1, W_1$ by minimizing $\mathcal{L}_{ce}$ on $\mathcal{D}_1$.
3: $\mathcal{P}_1 \leftarrow$ CalculatePrototypes($f_{\theta_1}, \mathcal{D}_1$).
4: **for** $t = 2, \ldots, T$ **do**
5:     **// Task t: Incremental Learning**
6:     $\mathcal{S}_{t-1} \leftarrow$ ComputeStableSubspace($f_{\theta_{t-1}}, W_{t-1}, \mathcal{P}_{t-1}$).
7:     Initialize $\theta_t \leftarrow \theta_{t-1}, W_t$ randomly.
8:     Update $\theta_t, W_t$ by minimizing $\mathcal{L}_{adapt}$ on $\mathcal{D}_t$ (Eq. 9).
9:     $\mathcal{P}_{aligned} \leftarrow$ AlignPrototypes($\mathcal{P}_{t-1}, \mathcal{S}_{t-1}, f_{\theta_t}, f_{\theta_{t-1}}$) (Eq. 11).
10:     $\mathcal{P}_{new} \leftarrow$ CalculatePrototypes($f_{\theta_t}, \mathcal{D}_t$).
11:     $\mathcal{P}_t \leftarrow \mathcal{P}_{aligned} \cup \mathcal{P}_{new}$.
12:     Recalibrate $W_t$ by minimizing $\mathcal{L}_{calib}$ on $\mathcal{P}_t$ and $\mathcal{D}_t$ (Eq. 12).
13: **end for**

---

### 3.5 SUBSPACE-GUIDED PROTOTYPE ALIGNMENT (SGPA)

Feature extractor updates cause past class prototypes $\{\mathbf{p}_i^{t-1}\}_{i=1}^{c_{t-1}}$ to become misaligned. SGPA addresses this by first aligning the prototypes(subspace-guided) and then calibrating the classifier.

**Stability-Modulated Prototype Alignment.** We correct prototype drift based on each prototype's stability. First, the average feature drift, $\Delta\mathbf{p}_{drift}$, is estimated from the current task's data: $\Delta\mathbf{p}_{drift} = \mathbb{E}_{\mathbf{x} \sim \mathcal{D}_t}[f_{\theta_t}(\mathbf{x}) - f_{\theta_{t-1}}(\mathbf{x})]$. Then, we compute a stability score $S_i$ for each prototype, defined as its normalized projection magnitude onto the stable subspace $\mathcal{S}$:

$$S_i = \frac{\|\text{Proj}_\mathcal{S}(\mathbf{p}_{t-1}^i)\|_2^2}{\|\mathbf{p}_{t-1}^i\|_2^2} = \frac{\sum_{j=1}^r (\mathbf{p}_{t-1}^i \cdot \mathbf{u}_j)^2}{\|\mathbf{p}_{t-1}^i\|_2^2}. \tag{10}$$

High $S_i$ indicates the prototype lies in high-curvature directions critical for past tasks. The alignment scales drift by plasticity $(1 - S_i)$:

$$\mathbf{p}_t^i = \mathbf{p}_{t-1}^i + (1 - S_i) \cdot \Delta\mathbf{p}_{drift}. \tag{11}$$

**Classifier Calibration.** With aligned prototypes $\{\mathbf{p}_t^i\}$, we retrain the classifier head. To prevent catastrophic forgetting, we generate synthetic features for past classes by sampling from Gaussian distributions centered at these aligned prototypes. Let $\mathcal{P}_{old} = \{(\mathbf{p}_t^i, y_i)\}_{i=1}^{c_{t-1}}$ be the set of aligned prototypes and their labels for past classes. The calibration loss is:

$$\mathcal{L}_{calib} = \mathbb{E}_{(\mathbf{x},y) \sim \mathcal{D}_t} \left[ \mathcal{L}_{ce}(W_t f_{\theta_t}(\mathbf{x}), y) \right] + \mathbb{E}_{\substack{\mathbf{p}_t^i, y_i \sim \mathcal{P}_{old} \\ \hat{\mathbf{z}} \sim \mathcal{N}(\mathbf{p}_t^i, \boldsymbol{\Sigma}_i)}} \left[ \mathcal{L}_{ce}(W_t \hat{\mathbf{z}}, y_i) \right], \tag{12}$$

where $\boldsymbol{\Sigma}_i$ is the covariance matrix of class $i$, computed from the features of class $i$'s original training samples under the updated feature extractor $f_{\theta_t}$: $\boldsymbol{\Sigma}_i = \frac{1}{N_i-1} \sum_{j=1}^{N_i} (f_{\theta_t}(\mathbf{x}_j^{(i)}) - \mathbf{p}_t^i)(f_{\theta_t}(\mathbf{x}_j^{(i)}) - \mathbf{p}_t^i)^\top$, where $\{\mathbf{x}_j^{(i)}\}_{j=1}^{N_i}$ are the $N_i$ training samples of class $i$. This ensures the classifier is properly adapted to the updated feature space for all classes.

## 4 EXPERIMENTAL RESULTS

In this section, we first present the experimental setups, compare SGCL's performance against state-of-the-art EFCIL methods, and finally provide a detailed analysis to validate our approach.

### 4.1 EXPERIMENTAL SETTINGS

**Datasets and Metrics** We evaluate our method on three standard benchmarks: CIFAR-100 (Krizhevsky & Hinton, 2009), Tiny-ImageNet (Wu et al., 2017), and ImageNet-Subset (Deng et al., 2009). Following the Cold Start EFCIL protocol, classes are split uniformly across tasks. For CIFAR-100 and ImageNet-Subset, we use configurations of 10 tasks with 10 classes each and 20

tasks with 5 classes each. For Tiny-ImageNet, we use 10 tasks with 20 classes and 20 tasks with 10 classes. We report two primary metrics: (1) **Average Accuracy (Acc)**, also known as Last Accuracy, measures the average accuracy on all seen classes after the final task; (2) **Average Anytime Accuracy (AAA)**, equivalent to Average Incremental Accuracy, evaluates the average performance throughout the entire learning process. Specifically, for CIFAR-100 and Tiny-ImageNet, each class contains 500 training samples, with 100 and 50 test samples respectively. For ImageNet-Subset, we follow the protocol established by Douillard et al. (2020), sampling 100 classes from ImageNet-1K, where each class contains approximately 1,300 training samples and 50 test samples.

**Competing Methods and Implementation Details.** We compare SGCL against a comprehensive set of baselines covering classic regularization (EWC, LwF), feature-space management (PASS, SSRE, EFC), and modern prototype-based methods (FeTrIL, LDC, ADC). Replay-based methods are excluded as they do not fit the EFCIL problem definition. For all experiments, we use a ResNet-18 backbone with a batch size of 64. The first task is trained for 100 epochs (160 for ImageNet-Subset), and subsequent tasks for 100 epochs. For CIFAR-100 and Tiny-ImageNet, we use an SGD optimizer with momentum 0.9; the learning rate is 0.1 for the first task and 0.005 for subsequent tasks. For ImageNet-Subset, the first task is trained with SGD (learning rate 0.1, momentum 0.9), while subsequent tasks use an Adam optimizer with learning rates of $1 \times 10^{-5}$ for the backbone and $1 \times 10^{-4}$ for the classifier, with a weight decay of $5 \times 10^{-4}$. For classifier calibration, we train the classifier for 30 epochs on both prototype and current task features using SGD with a learning rate of $1 \times 10^{-3}$, and batch size 256. The regularization weights $(\lambda_s, \lambda_p)$ are set to $(5, 0.03)$, $(10, 0.03)$, and $(20, 0.1)$ for CIFAR-100, Tiny-ImageNet, and ImageNet-Subset, respectively. All experiments were conducted on a single NVIDIA RTX 3090Ti GPU.

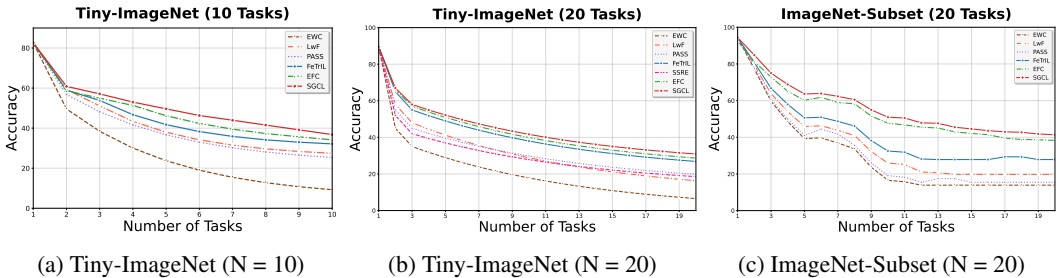

|  (a) Tiny-ImageNet (N = 10)  |  (b) Tiny-ImageNet (N = 20)  |  (c) ImageNet-Subset (N = 20)  |

Figure 3: Performance comparison during sequential training on (a) Tiny-ImageNet (N = 10), (b) Tiny-ImageNet (N = 20), and (c) ImageNet-Subset (N = 20). The figures show the average accuracy evolution across different continual learning methods.

## 4.2 COMPARISON WITH STATE-OF-THE-ART

We compare SGCL against a range of strong EFCIL baselines, including classic regularization methods like EWC (Kirkpatrick et al., 2017) and LwF (Hou et al., 2019), as well as recent state-of-the-art approaches such as PASS (Zhu et al., 2021b), FeTrIL (Petit et al., 2023), SSRE (Zhu et al., 2022), EFC (Magistri et al., 2024), ADC (Goswami et al., 2024), LDC (Gomez-Villa et al., 2024), DPCR (He et al., 2025). The comprehensive results, detailed in Table 1 and Figure 3, show that SGCL achieves highly competitive performance across diverse scenarios. On the standard 10-task splits for CIFAR-100, Tiny-ImageNet, and ImageNet-Subset, our method achieves the best final accuracies of 49.68%, 36.78%, and 53.52%, respectively. Notably, SGCL wins in 11 out of 12 settings, demonstrating consistent superiority. The robustness of our approach is particularly evident in the challenging 20-task settings, where SGCL achieves the best performance on ImageNet-Subset (41.44%), significantly outperforming the second-best method, DPCR (36.06%). While DPCR achieves slightly higher accuracy on CIFAR-100 20-task (37.98% vs. 37.23%), SGCL maintains better overall performance as measured by AAA (49.80% vs. 49.77%). These results validate the effectiveness of our subspace decomposition method in achieving an excellent stability-plasticity balance.

Table 1: Performance comparison of SGCL against other EFCIL methods across CIFAR-100, Tiny-ImageNet, and ImageNet-Subset under 10-task and 20-task configurations. We report the average accuracy (Acc) and average anytime accuracy (AAA) in percent (%). The best results are **bolded**, and second-best results are underlined.

| Method | CIFAR-100 | | | | Tiny-ImageNet | | | | ImageNet-Subset | | | |
|---|---|---|---|---|---|---|---|---|---|---|---|---|
| | 10 Tasks | | 20 Tasks | | 10 Tasks | | 20 Tasks | | 10 Tasks | | 20 Tasks | |
| | Acc | AAA | Acc | AAA | Acc | AAA | Acc | AAA | Acc | AAA | Acc | AAA |
| EWC | 32.35 | 49.14 | 18.72 | 31.02 | 9.25 | 24.01 | 6.55 | 15.70 | 25.90 | 39.40 | 13.89 | 26.95 |
| LwF | 33.95 | 55.20 | 18.75 | 38.39 | 27.45 | 45.14 | 16.30 | 32.94 | 38.95 | 56.41 | 19.75 | 40.23 |
| PASS | 31.75 | 47.86 | 18.65 | 32.86 | 25.35 | 39.25 | 19.85 | 32.01 | 27.65 | 45.74 | 15.45 | 31.65 |
| FeTrIL | 35.80 | 51.20 | 24.50 | 38.48 | 32.15 | 45.60 | 26.85 | 39.54 | 37.35 | 52.63 | 27.85 | 42.43 |
| SSRE | 31.65 | 47.26 | 18.75 | 32.45 | 24.15 | 38.82 | 18.55 | 30.62 | 26.65 | 43.76 | 17.45 | 31.15 |
| EFC | 43.95 | 58.58 | 32.15 | 47.70 | 34.45 | 47.95 | 28.69 | 42.07 | 47.38 | 60.30 | 35.75 | 49.92 |
| ADC | 46.48 | 61.35 | 35.13 | 47.56 | 32.32 | 43.04 | 21.33 | 37.80 | 46.58 | 67.07 | 30.83 | 49.23 |
| LDC | 45.40 | 59.50 | 36.85 | 48.87 | 34.20 | 46.80 | 24.95 | 40.33 | 51.40 | 69.40 | 31.52 | 50.60 |
| DPCR | 49.58 | 62.86 | **37.98** | 49.77 | 35.01 | 47.48 | 26.85 | 38.65 | 49.94 | 67.23 | 36.06 | 51.29 |
| SGCL | **49.68** | **62.88** | 37.23 | **49.80** | **36.78** | **48.72** | **30.92** | **45.51** | **53.52** | **69.54** | **41.44** | **55.60** |

## 4.3 ABLATION STUDY

To validate the effectiveness of our proposed components, we conduct comprehensive ablation studies on different datasets. Figure 4 shows the results of our ablation studies on CIFAR-100 and Tiny-ImageNet with 20 tasks each.

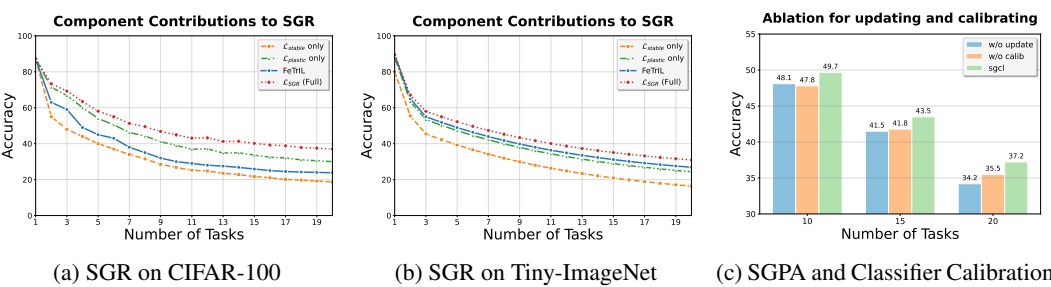

(a) SGR on CIFAR-100          (b) SGR on Tiny-ImageNet          (c) SGPA and Classifier Calibration

Figure 4: Ablation study of SGCL components on CIFAR-100 and Tiny-ImageNet (20 tasks).

**Components of SGR.** To isolate the contribution of our Subspace-Guided Regularization, we integrated it into a strong baseline, FeTrIL (Petit et al., 2023), and conducted ablation studies on CIFAR-100 and Tiny-ImageNet (Figures 4a and 4b). We compare the baseline against variants with only stable regularization ($\mathcal{L}_{stable}$), only plastic regularization ($\mathcal{L}_{plastic}$), and the full SGR. As shown in the figures, applying only the plastic regularization term yields a greater performance uplift over the baseline compared to applying only the stable term. This is consistent with the fact that the plastic subspace has a much higher dimensionality ($\geq d - c_{t-1} + 1$) than the stable subspace ($\leq c_{t-1} - 1$), giving it a larger influence on the feature space. However, the complete SGR model, which combines both terms, achieves the best performance, consistently outperforming both the baseline and the partial variants across both datasets. This confirms that both components are necessary and that their combination effectively balances stability and plasticity.

**Impact of SGPA and Classifier Calibration.** We validate the contributions of Subspace-Guided Prototype Alignment (SGPA) and classifier calibration in Figure 4c, comparing our full model against versions without prototype alignment (w/o update) and without calibration (w/o calib). Results show that removing either component degrades performance, confirming both are indispensable. The degradation from omitting SGPA is particularly pronounced in longer task sequences.

This is because uncorrected prototype drift accumulates; these increasingly erroneous prototypes are then fed into the calibration step, further corrupting the classifier and exacerbating forgetting.

### 4.4 ANALYSIS

**Visualization of Subspace-Guided Drift Control** To directly validate our central hypothesis, we visualize the components of feature drift. We measure the drift for data from the initial task ($\mathcal{T}_0$) after training on all subsequent tasks on CIFAR-100 (10 tasks). This drift vector, $\Delta \mathbf{z} = f_{\theta_9}(\mathbf{x}) - f_{\theta_0}(\mathbf{x})$, is projected onto the stable subspace ($\mathcal{S}$) defined after $\mathcal{T}_0$ and its plastic complement. Figure 5a compares the magnitude distribution of these projections for SGCL under several hyperparameter settings against standard Feature Distillation (FD), a baseline method that applies a uniform penalty across all feature dimensions. The visualization provides a striking confirmation of our method's mechanism. Under FD, the drift is isotropic, scattered around the $y = x$ axis, as its uniform penalty does not distinguish between feature directions. In stark contrast, all variants of SGCL force the drift to be highly anisotropic, confining changes almost exclusively to the plastic subspace while keeping the stable components nearly unchanged. For instance, increasing the stability regularization $\lambda_s$ (from 5 to 10) further reduces drift in the stable subspace, while decreasing the plasticity regularization $\lambda_p$ (from 0.03 to 0.02) allows for greater changes in the plastic subspace. This directly demonstrates that SGR successfully and controllably protects knowledge of past tasks in a targeted manner.

**Plasticity and Stability Analysis** Our method uses hyperparameters $\lambda_p$ and $\lambda_s$ to independently control plasticity and stability, demonstrating a clear trade-off on CIFAR-100 (20 tasks). Plasticity is governed by $\lambda_p$ through regularization of the plastic subspace; a smaller value allows for better adaptation and higher current-task accuracy (Figure 5c). Stability is managed by $\lambda_s$, which penalizes drift in the stable subspace. We measure knowledge retention using the average forgetting rate—the average accuracy drop on past tasks after training on a new one. A stronger penalty with a higher $\lambda_s$ mitigates catastrophic forgetting, reflected in a lower average forgetting rate (Figure 5b). These results confirm that our parameters provide a structured mechanism to navigate the stability-plasticity dilemma.

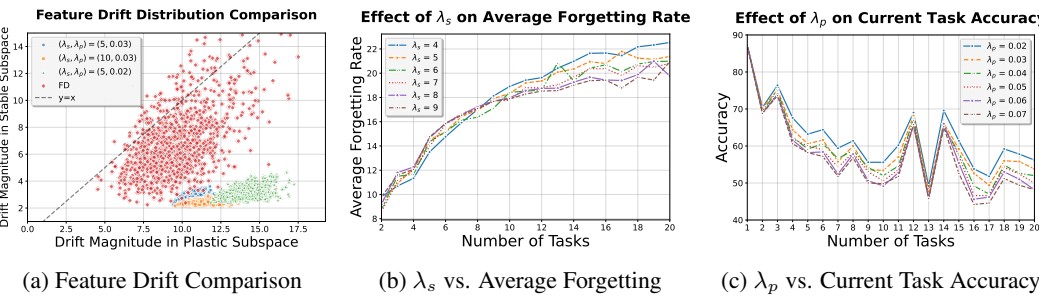

(a) Feature Drift Comparison  (b) $\lambda_s$ vs. Average Forgetting  (c) $\lambda_p$ vs. Current Task Accuracy

Figure 5: Analysis of feature drift and hyperparameters: (a) Feature drift comparison between methods, (b) Effect of $\lambda_s$ on stability, and (c) Effect of $\lambda_p$ on plasticity.

**Computational Efficiency** We analyze the computational and memory (RAM) overhead of our regularization term against EFC (Magistri et al., 2024), a highly efficient feature-space regularization method. Over the 20-task learning process, SGCL shows substantial gains (Table 2), reducing total GFLOPS by $3.7\times$ to $7.5\times$ and requiring significantly less peak memory (e.g., 0.19 MB vs. 44.6 MB on CIFAR-100). This advantage stems from our efficient subspace identification algorithm, which operates on a much smaller reduced matrix, avoiding the large $d \times d$ matrices required by EFC.

## 5 CONCLUSIONS AND LIMITATIONS

In this paper, we introduced Subspace-Guided Continual Learning (SGCL), a simple and effective method for Exemplar-Free Class-Incremental Learning. Our approach reframes the stability-plasticity dilemma by decomposing the feature space into a stable subspace derived from the feature-space Hessian and its plastic complement. This principled and computationally efficient strategy allows for selective knowledge preservation while accommodating new learning, leading to highly

| Dataset | Total GFLOPS | | Peak Memory (MB) | |
|---|---|---|---|---|
| | EFC | SGCL | EFC | SGCL |
| CIFAR-100 | 1,870.32 | 249.04 | 44.6 | 0.19 |
| Tiny-ImageNet | 3,693.89 | 983.70 | 44.6 | 0.38 |
| ImageNet-Subset | 4,769.32 | 635.04 | 44.6 | 0.19 |

Table 2: Total computational cost (GFLOPS) and peak memory (RAM) usage (MB) for the regularization term over the entire 20-task training process.

competitive performance across various benchmarks. Despite its effectiveness, SGCL has limitations that point to promising directions for future research. Its performance is sensitive to the hyperparameters $\lambda_s$ and $\lambda_p$, suggesting a need for automated tuning mechanisms. Moreover, the stable subspace is computed statically at the end of each task, which may become suboptimal as the feature extractor evolves. A critical research direction is therefore to develop techniques that can efficiently update this subspace dynamically during training, allowing it to co-evolve with the feature representation for a more precise stability-plasticity trade-off.

## ETHICS STATEMENT

This research focuses on the fundamental machine learning problem of continual learning, aiming to improve the stability and efficiency of AI models. All experiments were conducted on publicly available and widely used academic benchmark datasets (CIFAR-100, Tiny-ImageNet, and ImageNet-Subset), which do not contain personally identifiable information or other sensitive content. Our work does not involve human subjects, and we foresee no direct negative societal impacts or ethical concerns arising from our method or experiments. We are committed to the responsible development of AI, and we believe that advancing the robustness of learning systems is a crucial step toward creating more reliable and safe AI applications.

## REPRODUCIBILITY STATEMENT

We are committed to ensuring the reproducibility of our research. To this end, we have provided a detailed description of our proposed method, Subspace-Guided Continual Learning (SGCL), including all necessary mathematical formulations and a step-by-step pseudo-code in Algorithm 1. Our experimental setup, including the datasets, evaluation protocols, and metrics, is thoroughly described in Section 4.1. We have also provided comprehensive implementation details, such as the network architecture (ResNet-18), batch sizes, optimizers, learning rates, and all model-specific hyperparameters for each dataset, to allow for a faithful reimplementation of our experiments. To further facilitate verification and future research, we will make our source code and experiment configurations publicly available upon the paper's acceptance.

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

## LLM Usage Section

During the preparation of this manuscript, we utilized a large language model (LLM) to assist with improving the clarity, grammar, and overall readability of the text. The LLM's role was strictly limited to that of a writing assistant for language enhancement. For example, sections such as the abstract were refined with the help of the LLM to ensure the terminology is precise and the key contributions are communicated effectively. All scientific contributions, including the core ideas, methodology, experimental design, and analysis, were conceived and executed solely by the authors. The final version of the manuscript was thoroughly reviewed and edited by the authors to ensure its accuracy and integrity.

## A  Derivation of the Feature-Space Hessian

We derive the analytical form of the feature-space Hessian for the cross-entropy loss. Let $\mathcal{L}_{ce}$ be the cross-entropy loss for a single sample $(\mathbf{x}, y)$, where $y$ is the ground-truth label. The model consists of a feature extractor $\mathbf{z} = f_\theta(\mathbf{x})$ and a linear classifier $W \in \mathbb{R}^{c \times d}$.

The logits are given by $\mathbf{a} = W\mathbf{z}$. The predicted probabilities are computed using the softmax function, $\mathbf{p} = \text{softmax}(\mathbf{a})$, where $p_i = \frac{e^{a_i}}{\sum_{j=1}^{c} e^{a_j}}$. The cross-entropy loss is then:

$$\mathcal{L}_{ce} = -\log p_y = -a_y + \log \left( \sum_{j=1}^{c} e^{a_j} \right). \tag{13}$$

We are interested in the Hessian of this loss with respect to the feature vector $\mathbf{z}$, i.e., $\mathbf{H}_{\mathbf{z}} = \nabla_{\mathbf{z}}^2 \mathcal{L}_{ce} = \frac{\partial^2 \mathcal{L}_{ce}}{\partial \mathbf{z} \partial \mathbf{z}^\top}$.

First, we compute the gradient of the loss with respect to the logits $\mathbf{a}$. For the $i$-th component of the logits, we have:

$$\frac{\partial \mathcal{L}_{\text{ce}}}{\partial a_i} = \frac{\partial}{\partial a_i}\left(-a_y + \log\left(\sum_{j=1}^{c} e^{a_j}\right)\right) = -\delta_{iy} + \frac{e^{a_i}}{\sum_{j=1}^{c} e^{a_j}} = p_i - \delta_{iy}, \tag{14}$$

where $\delta_{iy}$ is the Kronecker delta. In vector form, this is $\nabla_{\mathbf{a}}\mathcal{L}_{\text{ce}} = \mathbf{p} - \mathbf{e}_y$, where $\mathbf{e}_y$ is the one-hot vector for the label $y$.

Next, we apply the chain rule to find the gradient with respect to the features $\mathbf{z}$:

$$\nabla_{\mathbf{z}}\mathcal{L}_{\text{ce}} = \frac{\partial \mathbf{a}^\top}{\partial \mathbf{z}}\nabla_{\mathbf{a}}\mathcal{L}_{\text{ce}} = W^\top(\mathbf{p} - \mathbf{e}_y). \tag{15}$$

To compute the Hessian, we differentiate the gradient $\nabla_{\mathbf{z}}\mathcal{L}_{\text{ce}}$ with respect to $\mathbf{z}^\top$:

$$\mathbf{H_z} = \frac{\partial}{\partial \mathbf{z}^\top}\left(W^\top(\mathbf{p} - \mathbf{e}_y)\right) = W^\top \frac{\partial \mathbf{p}}{\partial \mathbf{z}^\top}. \tag{16}$$

We need the Jacobian of the probability vector $\mathbf{p}$ with respect to the features $\mathbf{z}$. Using the chain rule again:

$$\frac{\partial \mathbf{p}}{\partial \mathbf{z}^\top} = \frac{\partial \mathbf{p}}{\partial \mathbf{a}^\top}\frac{\partial \mathbf{a}}{\partial \mathbf{z}^\top}. \tag{17}$$

The Jacobian of the logits $\mathbf{a} = W\mathbf{z}$ with respect to $\mathbf{z}$ is simply $\frac{\partial \mathbf{a}}{\partial \mathbf{z}^\top} = W$. The Jacobian of the softmax function is a standard result:

$$\frac{\partial p_i}{\partial a_j} = p_i(\delta_{ij} - p_j). \tag{18}$$

In matrix form, this Jacobian is $\frac{\partial \mathbf{p}}{\partial \mathbf{a}^\top} = \text{diag}(\mathbf{p}) - \mathbf{p}\mathbf{p}^\top$.

Combining these results, we get:

$$\frac{\partial \mathbf{p}}{\partial \mathbf{z}^\top} = (\text{diag}(\mathbf{p}) - \mathbf{p}\mathbf{p}^\top)W. \tag{19}$$

Substituting this back into the expression for the Hessian $\mathbf{H_z}$ gives the Hessian for a single sample:

$$\mathbf{H_z} = W^\top(\text{diag}(\mathbf{p}) - \mathbf{p}\mathbf{p}^\top)W. \tag{20}$$

The full feature-space Hessian $\mathbf{H}_f$ is the expectation of this quantity over the data distribution $\mathcal{D}_{t-1}$, which gives the expression in Equation 4:

$$\mathbf{H}_f = \mathbb{E}_{\mathbf{x}\sim\mathcal{D}_{t-1}}\left[W_{t-1}^\top(\text{diag}(\mathbf{p}) - \mathbf{p}\mathbf{p}^\top)W_{t-1}\right]. \tag{21}$$

# B PROOF OF PROPOSITIONS

## B.1 PROOF OF PROPOSITION 1

*Proof.* We provide an intuitive, step-by-step argument that makes explicit: (i) the softmax/logit-shift invariance, (ii) the covariance interpretation of the logits Hessian, and (iii) how rank transfers through the linear map induced by $W$.

**Step 1: Single-sample logits Hessian is a covariance.** For a fixed input $\mathbf{x}$, let $\mathbf{a} = W\mathbf{z}$ denote the logits and $\mathbf{p} = \text{softmax}(\mathbf{a}) \in (0,1)^c$ with $\sum_i p_i = 1$. The Hessian of the cross-entropy loss w.r.t. logits is

$$\mathbf{H}_a(\mathbf{x}) = \nabla_{\mathbf{a}}^2 \mathcal{L}_{\text{ce}} = \text{diag}(\mathbf{p}) - \mathbf{p}\mathbf{p}^\top. \tag{22}$$

Consider the random one-hot vector $\mathbf{Y} \in \{\mathbf{e}_1, \ldots, \mathbf{e}_c\}$ drawn from $\mathbb{P}(\mathbf{Y} = \mathbf{e}_i) = p_i$. Then

$$\text{Cov}(\mathbf{Y}) = \mathbb{E}[\mathbf{Y}\mathbf{Y}^\top] - \mathbb{E}[\mathbf{Y}]\,\mathbb{E}[\mathbf{Y}]^\top = \text{diag}(\mathbf{p}) - \mathbf{p}\mathbf{p}^\top = \mathbf{H}_a(\mathbf{x}). \tag{23}$$

Hence for any $\mathbf{u} \in \mathbb{R}^c$,

$$\mathbf{u}^\top \mathbf{H}_a(\mathbf{x})\,\mathbf{u} = \text{Var}(\mathbf{u}^\top \mathbf{Y}) = \sum_{i=1}^{c} p_i u_i^2 - \left(\sum_{i=1}^{c} p_i u_i\right)^2 \geq 0, \tag{24}$$

with equality iff $\mathbf{u}^\top \mathbf{Y}$ is almost surely constant under strictly positive $\mathbf{p}$, which occurs exactly when $u_1 = \cdots = u_c$. Therefore, $\mathbf{H}_a(\mathbf{x})$ is positive semidefinite, its nullspace is $\mathrm{span}(\mathbf{1})$, and it is strictly positive definite on the subspace $\mathbf{1}^\perp = \{\mathbf{u} : \mathbf{1}^\top \mathbf{u} = 0\}$.

**Softmax/logit-shift invariance.** Adding any constant $\alpha$ to all logits leaves softmax unchanged: $\mathrm{softmax}(\mathbf{a} + \alpha\mathbf{1}) = \mathrm{softmax}(\mathbf{a})$. Directions along $\mathrm{span}(\mathbf{1})$ therefore produce no change in probabilities and incur zero curvature, matching the nullspace characterization above.

**Step 2: Expectation preserves structure and fixes rank.** Define the expected logits Hessian

$$\bar{\mathbf{H}}_a := \mathbb{E}_{\mathbf{x}}\left[\mathbf{H}_a(\mathbf{x})\right] = \mathbb{E}_{\mathbf{x}}\left[\mathrm{diag}(\mathbf{p}) - \mathbf{p}\mathbf{p}^\top\right]. \tag{25}$$

As a convex combination of PSD matrices, $\bar{\mathbf{H}}_a$ is PSD. The nullspace and positivity on $\mathbf{1}^\perp$ are preserved: for any $\mathbf{u}$,

$$\mathbf{u}^\top \bar{\mathbf{H}}_a \mathbf{u} = \mathbb{E}_{\mathbf{x}}\left[\mathbf{u}^\top \mathbf{H}_a(\mathbf{x})\mathbf{u}\right] = \mathbb{E}_{\mathbf{x}}\left[\mathrm{Var}_{i\sim\mathbf{p}(\mathbf{x})}(u_i)\right]. \tag{26}$$

This equals 0 iff $u_1 = \cdots = u_c$, hence $\mathrm{Null}(\bar{\mathbf{H}}_a) = \mathrm{span}(\mathbf{1})$ and $\bar{\mathbf{H}}_a \succ 0$ on $\mathbf{1}^\perp$. Consequently, $\mathrm{rank}(\bar{\mathbf{H}}_a) = c - 1$.

**Step 3: Transfer to feature space via $W$.** The expected feature-space Hessian is

$$\mathbf{H}_f = \mathbb{E}_{\mathbf{x}}\left[\nabla_{\mathbf{z}}^2 \mathcal{L}_{\mathrm{ce}}\right]_{\text{to features}} = W^\top \bar{\mathbf{H}}_a W. \tag{27}$$

For any $\mathbf{v} \in \mathbb{R}^d$,

$$\mathbf{v}^\top \mathbf{H}_f \mathbf{v} = (W\mathbf{v})^\top \bar{\mathbf{H}}_a (W\mathbf{v}) \geq 0. \tag{28}$$

Moreover,

$$\mathbf{v}^\top \mathbf{H}_f \mathbf{v} = 0 \iff W\mathbf{v} \in \mathrm{Null}(\bar{\mathbf{H}}_a) = \mathrm{span}(\mathbf{1}). \tag{29}$$

Intuitively, $\mathbf{v}$ is a feature direction that only adds an equal shift to all logits; cross-entropy (via softmax) is blind to such shifts, so curvature is zero along these directions.

**Upper bound on rank.** Using the property $\mathrm{rank}(\mathbf{A}^\top \mathbf{A}) \leq \mathrm{rank}(\mathbf{A})$ for any matrix $\mathbf{A}$, we have:

$$\mathrm{rank}(\mathbf{H}_f) = \mathrm{rank}(W^\top \bar{\mathbf{H}}_a W) \leq \min(\mathrm{rank}(W^\top), \mathrm{rank}(\bar{\mathbf{H}}_a)) = \min(\mathrm{rank}(W), c - 1). \tag{30}$$

**Tightness when $W$ has full row rank.** When $W \in \mathbb{R}^{c\times d}$ has full row rank (which implies $d \geq c$ and $\mathrm{rank}(W) = c$), its image is $\mathrm{Im}(W) = \mathbb{R}^c$, and $\mathrm{Im}(W^\top)$ is a $c$-dimensional subspace of $\mathbb{R}^d$. The kernel of $\bar{\mathbf{H}}_a^{1/2}$ is $\mathrm{span}(\mathbf{1})$, which is 1-dimensional. Since $\mathrm{Im}(W) = \mathbb{R}^c$ contains $\mathrm{span}(\mathbf{1})$, by the rank-nullity theorem for matrix products:

$$\mathrm{rank}(\bar{\mathbf{H}}_a^{1/2} W) = \mathrm{rank}(W) - \dim(\mathrm{Im}(W) \cap \mathrm{Ker}(\bar{\mathbf{H}}_a^{1/2})) = c - 1. \tag{31}$$

Therefore, when $W$ has full row rank, $\mathrm{rank}(\mathbf{H}_f) = c - 1$.

**Conclusion.** For general $W$, the rank of $\mathbf{H}_f$ is upper bounded by $\min(\mathrm{rank}(W), c - 1)$, and this bound is tight when $W$ has full row rank. The intrinsically flat direction corresponds to uniform logit shifts (softmax invariance), eliminating one degree of curvature from the logit subspace. $\qquad\square$

### B.2 PROOF OF PROPOSITION 2

*Proof.* We detail why the non-zero spectrum of $\mathbf{H}_f = W^\top \mathbf{A} W$ is identical to that of the reduced matrix $\mathbf{H}_{\mathrm{red}} = \mathbf{R}\mathbf{A}\mathbf{R}^\top$, and how this gives both correctness and efficiency.

**Step 1: Subspace structure via QR.** Take the thin QR of $W^\top$:

$$W^\top = \mathbf{Q}\mathbf{R}, \quad \mathbf{Q} \in \mathbb{R}^{d\times c}, \ \mathbf{Q}^\top \mathbf{Q} = \mathbf{I}_c, \ \mathbf{R} \in \mathbb{R}^{c\times c} \text{ upper triangular.} \tag{32}$$

The columns of $\mathbf{Q}$ form an orthonormal basis for $\mathrm{Im}(W^\top)$ (the row space of $W$). Using this factorization,

$$\mathbf{H}_f = W^\top \mathbf{A} W = (\mathbf{Q}\mathbf{R})\,\mathbf{A}\,(\mathbf{Q}\mathbf{R})^\top = \mathbf{Q}\,\underbrace{(\mathbf{R}\mathbf{A}\mathbf{R}^\top)}_{\mathbf{H}_{\mathrm{red}}}\,\mathbf{Q}^\top. \tag{33}$$

Hence $\mathbf{H}_f$ acts trivially (as zero) on the orthogonal complement $\mathrm{Im}(\mathbf{Q})^\perp$, and maps $\mathrm{Im}(\mathbf{Q})$ into itself via the $c \times c$ operator $\mathbf{H}_{\mathrm{red}}$ (expressed in the $\mathbf{Q}$-basis).

**Step 2: Eigenpair correspondence (both directions).** Any $\mathbf{v} \in \mathbb{R}^d$ decomposes uniquely as $\mathbf{v} = \mathbf{Q}\mathbf{u} + \mathbf{v}_\perp$ with $\mathbf{v}_\perp \perp \mathrm{Im}(\mathbf{Q})$. Since $\mathbf{Q}^\top \mathbf{v}_\perp = \mathbf{0}$, we have $\mathbf{H}_f \mathbf{v} = \mathbf{Q}\,\mathbf{H}_{\text{red}}\,\mathbf{u}$. Therefore, any eigenvector with non-zero eigenvalue must lie in $\mathrm{Im}(\mathbf{Q})$: if $\mathbf{H}_f \mathbf{v} = \sigma \mathbf{v}$ and $\sigma \neq 0$, then $\mathbf{v}_\perp = \mathbf{0}$ and $\mathbf{v} = \mathbf{Q}\mathbf{u}$.

Substituting $\mathbf{v} = \mathbf{Q}\mathbf{u}$ into $\mathbf{H}_f \mathbf{v} = \sigma \mathbf{v}$ gives

$$\mathbf{Q}\,\mathbf{H}_{\text{red}}\,\mathbf{u} = \sigma\,\mathbf{Q}\mathbf{u} \quad \Longleftrightarrow \quad \mathbf{H}_{\text{red}}\,\mathbf{u} = \sigma\,\mathbf{u}, \tag{34}$$

because $\mathbf{Q}$ has full column rank and $\mathbf{Q}^\top \mathbf{Q} = \mathbf{I}$. Thus, non-zero eigenpairs $(\sigma, \mathbf{v})$ of $\mathbf{H}_f$ correspond bijectively to eigenpairs $(\sigma, \mathbf{u})$ of $\mathbf{H}_{\text{red}}$ via $\mathbf{v} = \mathbf{Q}\mathbf{u}$; conversely, any eigenpair of $\mathbf{H}_{\text{red}}$ lifts to one of $\mathbf{H}_f$.

**Step 3: Rayleigh quotient equality (spectral identity).** For $\mathbf{v} = \mathbf{Q}\mathbf{u}$,

$$\frac{\mathbf{v}^\top \mathbf{H}_f \mathbf{v}}{\mathbf{v}^\top \mathbf{v}} = \frac{\mathbf{u}^\top \mathbf{H}_{\text{red}} \mathbf{u}}{\mathbf{u}^\top \mathbf{u}}, \tag{35}$$

since $\mathbf{v}^\top \mathbf{H}_f \mathbf{v} = \mathbf{u}^\top \mathbf{H}_{\text{red}} \mathbf{u}$ and $\mathbf{v}^\top \mathbf{v} = \mathbf{u}^\top \mathbf{u}$ by $\mathbf{Q}^\top \mathbf{Q} = \mathbf{I}$. This shows $\mathbf{H}_f$ and $\mathbf{H}_{\text{red}}$ have identical non-zero eigenvalues (same extremal Rayleigh quotients) and identical inertia on $\mathrm{Im}(\mathbf{Q})$.

**Step 4: Complexity implication.** Computing $\mathbf{Q}, \mathbf{R}$ costs $O(dc^2)$. Forming $\mathbf{H}_{\text{red}} = \mathbf{R}\mathbf{A}\mathbf{R}^\top$ costs $O(c^3)$ (two $c \times c$ multiplications). Eigendecomposition of $\mathbf{H}_{\text{red}}$ also costs $O(c^3)$. Thus overall $O(dc^2 + c^3)$, versus forming $\mathbf{H}_f$ explicitly ($O(d^2c)$) and decomposing it ($O(d^3)$) when $c \ll d$.

Therefore, the non-zero eigenvalues (and corresponding eigenvectors) are obtained equivalently and far more efficiently via the $c \times c$ reduced problem. $\qquad\square$

## C  FURTHER ANALYSIS

### C.1  ALTERNATIVE SUBSPACE IDENTIFICATION

Our core assumption is that the feature space can be functionally decomposed into stable and plastic subspaces. While we identify these subspaces via the feature-space Hessian, alternative identification strategies are possible. A natural candidate is Principal Component Analysis (PCA) based on within-class covariance matrices.

**PCA-based Subspace Identification.** Given the feature space $\mathbb{R}^d$ and $c_{t-1}$ learned classes, let $\Sigma_i \in \mathbb{R}^{d \times d}$ denote the covariance matrix of class $i$ with $n_i$ samples. The weighted average covariance matrix is:

$$\bar{\Sigma} = \sum_{i=1}^{c_{t-1}} \frac{n_i}{N} \Sigma_i, \quad N = \sum_{i=1}^{c_{t-1}} n_i. \tag{36}$$

Performing eigendecomposition $\bar{\Sigma} = U \Lambda U^T$ with $\Lambda = \mathrm{diag}(\lambda_1, \ldots, \lambda_d)$ and $\lambda_1 \geq \lambda_2 \geq \cdots \geq \lambda_d \geq 0$, the stable subspace is defined by the top $K = c_{t-1} - 1$ principal components:

$$\mathcal{S}_{\text{PCA}} = \mathrm{span}\{u_1, u_2, \ldots, u_K\}, \tag{37}$$

with the remaining $d - K$ dimensions forming the plastic subspace $\mathcal{P}_{\text{PCA}}$. This approach uses first-order statistics (covariance) with computational complexity $O(d^2c + d^3)$.

**Comparative Evaluation.** Table 3 compares the two identification strategies on CIFAR-100 and Tiny-ImageNet. For a fair comparison, both methods use only SGR for feature regularization, excluding SGPA prototype alignment and classifier calibration. Results show that the Hessian-based method consistently outperforms the PCA-based approach across all settings, with the advantage being particularly pronounced in the 20-task configurations. This validates that loss curvature analysis more precisely captures feature directions critical for preserving past knowledge.

### C.2  SGR AND SGPA COUPLING

In our work, we address feature drift and prototype drift within a unified framework. SGR constrains feature drift by penalizing changes in the stable subspace, while SGPA leverages the same subspace information to guide prototype alignment. This coupling is not only theoretically coherent

Table 3: Performance comparison of different subspace identification methods (SGR only, without SGPA and classifier calibration).

| Method | CIFAR-100 | | | | Tiny-ImageNet | | | |
|---|---|---|---|---|---|---|---|---|
| | 10 Tasks | | 20 Tasks | | 10 Tasks | | 20 Tasks | |
| | Acc | AAA | Acc | AAA | Acc | AAA | Acc | AAA |
| PCA-based | 36.50 | 48.68 | 20.50 | 32.21 | 31.59 | 41.84 | 23.22 | 36.12 |
| Hessian-based | **44.15** | **58.32** | **31.67** | **45.55** | **34.21** | **45.18** | **27.35** | **39.86** |

but also computationally efficient, as SGPA reuses the stable subspace computed for SGR without introducing extra overhead.

To demonstrate the effectiveness of this integrated design, we compare our full model against hybrid versions where SGR is paired with other general-purpose prototype drift compensation modules: SDC (Semantic Drift Compensation)(Yu et al., 2020), LDC (Learnable Drift Compensation)(Gomez-Villa et al., 2024), and ADC (Adversarial Drift Compensation)(Goswami et al., 2024). The results on CIFAR-100 are presented in Table 4.

Table 4: Performance comparison on CIFAR-100 (10/20 tasks) with SGR paired with different drift-compensation modules. Our coupled SGPA approach yields the best overall performance.

| Method | 10 Tasks | | 20 Tasks | |
|---|---|---|---|---|
| | Acc | AAA | Acc | AAA |
| SGR + ADC | 43.00 | 59.05 | 32.76 | 45.08 |
| SGR + SDC | 48.98 | 62.12 | 36.55 | 48.56 |
| SGR + LDC | 49.10 | 62.21 | 35.04 | 48.28 |
| SGR + SGPA (Ours) | **49.68** | **62.88** | **37.23** | **49.80** |

As shown in the table, while SGR provides a strong foundation that improves performance with all drift compensation modules, the tightly coupled SGR+SGPA configuration achieves the best results. This highlights the benefit of our unified, co-designed approach to managing both feature and prototype drift.

### C.3 CHOICE OF $\lambda_s$ AND $\lambda_p$

**Theory-Guided Principles.** In our method, $\lambda_s$ and $\lambda_p$ are critical hyperparameters governing the trade-off between stability and plasticity. Although their optimal values depend on the specific dataset, their adjustment follows principled patterns derived from our method's geometric nature:

*Role of the hyperparameters.* To prioritize knowledge retention while enabling adaptation, we consistently enforce $\lambda_s \gg \lambda_p$, where $\lambda_s$ controls the stable subspace and $\lambda_p$ controls the plastic subspace.

*Why we need $\lambda_p > 0$.* The stable subspace, while critical, has a much lower dimensionality compared to the full feature space. Relying solely on stable subspace regularization is insufficient to fully counteract forgetting caused by general feature drift, as visualized in Figure 5a. Therefore, a mild constraint on the plastic subspace ($\lambda_p > 0$) is essential to prevent excessive deviation in the remaining dimensions. Unlike other methods (Magistri et al., 2024) that often add a separate global penalty term (e.g., $\|\Delta \mathbf{z}\|_2^2$), SGCL naturally integrates this control via $\lambda_p$ (see Table 5). Our ablation study in Figure 4a confirms that setting $\lambda_p = 0$ leads to significant performance degradation.

*Parameter sensitivity.* Since the stable subspace is low-dimensional, $\lambda_s$ is robust to large changes and can be adjusted over a wide range. Conversely, the high-dimensional plastic subspace exerts a strong influence on total loss, requiring finer tuning of $\lambda_p$. Table 6 demonstrates the robustness of $\lambda_s$ across challenging datasets, where performance remains stable across different values.

*Dataset difficulty.* Challenging datasets (e.g., Tiny-ImageNet, ImageNet-Subset) require stronger constraints on global feature drift $\Delta \mathbf{z}$. Consequently, both parameters should be increased relative

Table 5: Impact of different plasticity regularization terms on CIFAR-100 ($\lambda_s = 5$).

| Plasticity Term ($\lambda_p = 0.03$) | 10-task Acc | 20-task Acc |
|---|---|---|
| None ($\lambda_p = 0$) | 28.77 | 19.86 |
| Global Drift ($\|\Delta \mathbf{z}\|_2^2$) | 48.73 | 35.69 |
| Plastic Subspace (Ours) | **49.68** | **37.23** |

Table 6: Robustness of $\lambda_s$ on Tiny-ImageNet ($\lambda_p = 0.03$) and ImageNet-Subset ($\lambda_p = 0.1$).

| $\lambda_s$ | Tiny-ImageNet | | ImageNet-Subset | |
|---|---|---|---|---|
| | 10-Task | 20-Task | 10-Task | 20-Task |
| 10 | 36.78 | 30.92 | 46.58 | 33.21 |
| 11 | 36.10 | 30.03 | 47.14 | 34.65 |
| 12 | 35.41 | 29.24 | 48.08 | 35.11 |

to simpler baselines. Consistent with the sensitivity principle, $\lambda_s$ can be increased substantially, whereas $\lambda_p$ should be increased only slightly. Table 7 demonstrates this principle through a post-hoc analysis starting from the CIFAR-100 baseline, showing that both parameters need to be increased for more challenging datasets, with $\lambda_s$ adjustable in a wider range due to the low dimensionality of the stable subspace.

Table 7: Illustration of dataset difficulty principle: harder datasets require larger hyperparameters, starting from CIFAR-100 baseline ($\lambda_s = 5, \lambda_p = 0.03$).

| Dataset | ($\lambda_s, \lambda_p$) | Acc | Notes |
|---|---|---|---|
| Tiny-ImageNet (10 tasks) | (5, 0.03) | 34.91 | CIFAR-100 baseline |
| | (10, 0.03) | **36.78** | Increase $\lambda_s$ |
| | (5, 0.04) | 36.15 | Increase $\lambda_p$ a little |
| | (5, 0.05) | 34.80 | Increase more $\lambda_p$ |
| ImageNet-Subset (10 tasks) | (5, 0.03) | 47.22 | CIFAR-100 baseline |
| | (10, 0.05) | 51.63 | Moderate increase |
| | (20, 0.1) | **53.52** | Optimal setting |
| | (25, 0.15) | 49.87 | Over-regularization |

**Hyperparameter Search Strategy.** Following standard continual learning practices (Douillard et al., 2020), we randomly split each class's training data into 90% for training and 10% for validation. All hyperparameters were selected based on validation set performance, and final test results were obtained by retraining on the full training data.

We performed a limited, theory-guided search rather than exhaustive grid search. Starting with CIFAR-100 as the baseline, we searched $\lambda_s \in \{1, 5, 10\}$ and $\lambda_p \in \{0.02, 0.03\}$ to determine the optimal setting $(5, 0.03)$. For more challenging datasets, guided by the principles above, we progressively increased the search ranges: $\lambda_s \in \{8, 10\}$ and $\lambda_p \in \{0.03, 0.05\}$ for Tiny-ImageNet, and $\lambda_s \in \{10, 20\}$ and $\lambda_p \in \{0.05, 0.1\}$ for ImageNet-Subset. Importantly, hyperparameter search was conducted only on 10-task configurations, and the selected values were directly applied to 20-task experiments without further tuning.

### C.4 PERFORMANCE ON OTHER BENCHMARKS

Furthermore, we evaluate our method on other benchmarks, including CUB-200(Wah et al., 2011) and ImageNet-1K(Deng et al., 2009). The results are shown in Table 8. For ImageNet-1K, we maintain the same experimental settings as ImageNet-Subset. For CUB-200, to accommodate the smaller data scale, we reduce the backbone learning rate (e.g., $7 \times 10^{-4}$ for the initial task and $6 \times 10^{-5}$ subsequently), while keeping the regularization weights $\lambda_s = 20$ and $\lambda_p = 0.1$ consistent with ImageNet-Subset.

Table 8: Performance comparison on CUB-200 and ImageNet-1K benchmarks. We report Average Accuracy (Acc) and Average Anytime Accuracy (AAA).

| Method | CUB-200 | | | | ImageNet-1K | |
|--------|---------|---|---------|---|-------------|---|
| | 10 Tasks | | 20 Tasks | | 10 Tasks | |
| | Acc | AAA | Acc | AAA | Acc | AAA |
| EFC | 51.03 | 63.28 | 46.13 | 59.37 | 42.35 | 56.14 |
| ADC | 44.65 | 59.62 | 19.47 | 39.72 | 31.34 | 50.95 |
| LDC | 40.16 | 54.73 | 24.49 | 42.67 | 35.15 | 53.88 |
| **SGCL (Ours)** | **59.38** | **67.22** | **57.86** | **66.76** | **44.82** | **58.49** |

# D  THEORETICAL ANALYSIS UNDER THE SGR CONSTRAINT

In this appendix, we analyze SGCL under a local quadratic model with the subspace-guided regularization (SGR) used in the main method. We show that, once the feature-space Hessian $\mathbf{H}_f^{(t-1)}$ is fixed, choosing the stable subspace as the image of $\mathbf{H}_f^{(t-1)}$ (i.e., the span of all eigenvectors with non-zero eigenvalues) minimizes natural forgetting criteria among all subspace choices that are aligned with the eigenvectors of $\mathbf{H}_f^{(t-1)}$ and have the same dimension. This corresponds exactly to the Hessian-guided stable–plastic decomposition used in SGCL.

**Remark on SGR formulation:** For analytical tractability, the following theoretical analysis adopts an unweighted SGR formulation (Eq. 40), which applies uniform penalties $\lambda_s$ and $\lambda_p$ to the stable and plastic subspaces, respectively. In the actual implementation (Section 3.4), we use a curvature-weighted version (Eq. 7) where penalties are scaled by the corresponding eigenvalues $\sigma_i$ within the stable subspace. Both formulations lead to the same optimal subspace decomposition—namely, selecting all positive-curvature directions as the stable subspace—although the curvature-weighted version provides finer-grained control over forgetting in practice.

## D.1  LOCAL QUADRATIC MODEL AND SGR-INDUCED OBJECTIVE

Recall that the model consists of a feature extractor $f_\theta : \mathbb{R}^{d_{\mathrm{in}}} \to \mathbb{R}^d$ and a linear classifier $W_t \in \mathbb{R}^{c_t \times d}$, where $c_t = \sum_{k=1}^t |\mathcal{C}_k|$ is the total number of classes observed up to task $t$. For an input $\mathbf{x}$ we denote its feature by $\mathbf{z} = f_\theta(\mathbf{x}) \in \mathbb{R}^d$.

At the end of task $t - 1$, the parameters $(\theta_{t-1}, W_{t-1})$ are (approximately) a stationary point of the old-task loss $\mathcal{L}_{\leq t-1}$, which can be written as an average over the past task distributions $\{\mathcal{D}_k\}_{k=1}^{t-1}$ as in Eq. (1) of the main paper. Treating $W_{t-1}$ as fixed and viewing $\mathcal{L}_{\leq t-1}$ as a function of the feature representation $\mathbf{z}$, we denote the corresponding feature-space Hessian at task $t - 1$ by

$$\mathbf{H}_f^{(t-1)} \;=\; \nabla_{\mathbf{z}}^2 \mathcal{L}_{\leq t-1}(\mathbf{z})\Big|_{\mathbf{z}=\mathbf{z}_{t-1}}, \tag{38}$$

where $\mathbf{z}_{t-1} = f_{\theta_{t-1}}(\mathbf{x})$ is the feature before learning task $t$. As derived in the main paper (Eq. (4)), for the cross-entropy loss with a softmax classifier $\mathbf{H}_f^{(t-1)}$ is symmetric positive semidefinite.

For a small feature drift $\Delta\mathbf{z}_t \in \mathbb{R}^d$ induced by training task $t$, we adopt the second-order Taylor approximation of the old-task loss:

$$\Delta\mathcal{L}_{\leq t-1}^{(t)}(\Delta\mathbf{z}_t) \;:=\; \mathcal{L}_{\leq t-1}(\mathbf{z}_{t-1} + \Delta\mathbf{z}_t) - \mathcal{L}_{\leq t-1}(\mathbf{z}_{t-1}) \;\approx\; \frac{1}{2}\Delta\mathbf{z}_t^\top \mathbf{H}_f^{(t-1)}\Delta\mathbf{z}_t. \tag{39}$$

During task $t$, SGCL applies SGR on the feature drift by penalizing the components in the stable and plastic subspaces with different strengths. Let $\mathcal{S}_t \subset \mathbb{R}^d$ be a $k$-dimensional *stable* subspace and $\mathcal{P}_t = \mathcal{S}_t^\perp$ its orthogonal *plastic* complement. We denote by $P_{\mathcal{S}_t}$ and $P_{\mathcal{P}_t}$ the orthogonal projectors onto $\mathcal{S}_t$ and $\mathcal{P}_t$, respectively. Under a local approximation in feature space, the SGR constraint at task $t$ takes the quadratic form

$$\Omega_{\mathrm{SGR}}(\Delta\mathbf{z}_t; \mathcal{S}_t) \;=\; \lambda_s \|P_{\mathcal{S}_t}\Delta\mathbf{z}_t\|_2^2 \;+\; \lambda_p \|P_{\mathcal{P}_t}\Delta\mathbf{z}_t\|_2^2, \qquad \lambda_s > \lambda_p \geq 0, \tag{40}$$

where $\lambda_s$ and $\lambda_p$ are the stable and plastic SGR coefficients used in the main method.

Combining the new-task loss (locally linearized) with the quadratic approximation of the old-task loss and the SGR penalty yields the following local objective in feature space at task $t$:

$$\phi_t(\Delta\mathbf{z}_t; \mathcal{S}_t) \;=\; \mathbf{g}_t^\top \Delta\mathbf{z}_t \;+\; \frac{1}{2}\Delta\mathbf{z}_t^\top \mathbf{H}_f^{(t-1)}\Delta\mathbf{z}_t \;+\; \lambda_s\|P_{\mathcal{S}_t}\Delta\mathbf{z}_t\|_2^2 \;+\; \lambda_p\|P_{\mathcal{P}_t}\Delta\mathbf{z}_t\|_2^2, \tag{41}$$

where $\mathbf{g}_t$ is the gradient of the new-task loss with respect to $\mathbf{z}$, evaluated at $\mathbf{z}_{t-1}$. We refer to equation 41 as the SGR-induced local objective.

The minimizer of equation 41 with respect to $\Delta\mathbf{z}_t$ is the SGR-constrained local update in feature space at task $t$. Its form depends on the stable subspace $\mathcal{S}_t$. We now analyze how the resulting old-task loss increase in equation 39 depends on the choice of $\mathcal{S}_t$.

### D.2 DIAGONALIZATION IN THE HESSIAN EIGENBASIS

Let $\mathbf{H}_f^{(t-1)}$ admit the eigendecomposition

$$\mathbf{H}_f^{(t-1)} \;=\; \mathbf{U}\boldsymbol{\Sigma}\mathbf{U}^\top, \qquad \boldsymbol{\Sigma} \;=\; \mathrm{diag}(\sigma_1^{(t-1)},\ldots,\sigma_d^{(t-1)}), \tag{42}$$

where $\mathbf{U} = [\mathbf{u}_1^{(t-1)},\ldots,\mathbf{u}_d^{(t-1)}]$ is orthogonal and $\sigma_1^{(t-1)} \geq \cdots \geq \sigma_d^{(t-1)} \geq 0$ are the eigenvalues. Denote the rank of $\mathbf{H}_f^{(t-1)}$ by

$$r \;:=\; \mathrm{rank}\big(\mathbf{H}_f^{(t-1)}\big) \;=\; \big|\{i : \sigma_i^{(t-1)} > 0\}\big|. \tag{43}$$

In SGCL, the stable subspace is chosen as the image of $\mathbf{H}_f^{(t-1)}$, spanned by the eigenvectors with non-zero eigenvalues (see Sec. 3.2 of the main paper), while the plastic subspace is the orthogonal complement (the kernel of $\mathbf{H}_f^{(t-1)}$). Below we first analyze general choices of $k$-dimensional subspaces aligned with the eigenvectors, and then specialize to the case $k = r$ corresponding to SGCL.

Let $I_t \subset \{1,\ldots,d\}$ be an index set of size $k$. We define

$$\mathcal{S}_t(I_t) \;:=\; \mathrm{span}\{\mathbf{u}_i^{(t-1)} : i \in I_t\}, \qquad \mathcal{P}_t(I_t) \;:=\; \mathcal{S}_t(I_t)^\perp \;=\; \mathrm{span}\{\mathbf{u}_j^{(t-1)} : j \notin I_t\}. \tag{44}$$

Equivalently, for each eigendirection $\mathbf{u}_i^{(t-1)}$ we assign either the stable SGR coefficient $\lambda_s$ or the plastic SGR coefficient $\lambda_p$:

$$\lambda_i \;:=\; \begin{cases} \lambda_s, & i \in I_t, \\ \lambda_p, & i \notin I_t. \end{cases} \tag{45}$$

The choice of index set $I_t$ encodes the choice of stable subspace $\mathcal{S}_t$.

Writing $\mathbf{g}_t = \mathbf{U}\boldsymbol{\alpha}$ and $\Delta\mathbf{z}_t = \mathbf{U}\boldsymbol{\beta}$ in the eigenbasis, the SGR-induced objective equation 41 becomes

$$\phi_t(\Delta\mathbf{z}_t; \mathcal{S}_t(I_t)) \;=\; \sum_{i=1}^d \left(\alpha_i\beta_i + \frac{1}{2}\sigma_i^{(t-1)}\beta_i^2 + \lambda_i\beta_i^2\right), \tag{46}$$

where $\lambda_i$ is given by equation 45. The coordinates $\beta_i$ are decoupled, and minimizing equation 46 with respect to $\beta_i$ yields

$$\beta_i^\star \;=\; -\frac{\alpha_i}{\sigma_i^{(t-1)} + 2\lambda_i}, \qquad 1 \leq i \leq d. \tag{47}$$

Thus the SGR-constrained local feature update in the original space is

$$\Delta\mathbf{z}_t^\star(\mathcal{S}_t(I_t)) \;=\; \mathbf{U}\boldsymbol{\beta}^\star \;=\; -\sum_{i=1}^d \frac{\alpha_i}{\sigma_i^{(t-1)} + 2\lambda_i}\, \mathbf{u}_i^{(t-1)}. \tag{48}$$

Substituting equation 47 into equation 39, the induced increase in the old-task loss is

$$\Delta\mathcal{L}_{\leq t-1}^{(t)}\big(\Delta\mathbf{z}_t^\star(\mathcal{S}_t(I_t))\big) \;=\; \frac{1}{2}\sum_{i=1}^d \sigma_i^{(t-1)}\frac{\alpha_i^2}{\big(\sigma_i^{(t-1)} + 2\lambda_i\big)^2}. \tag{49}$$

We define the per-direction *SGR forgetting coefficients*

$$m_i(\lambda_i) := \frac{\sigma_i^{(t-1)}}{\left(\sigma_i^{(t-1)} + 2\lambda_i\right)^2}, \qquad 1 \le i \le d, \tag{50}$$

so that equation 49 can be written compactly as

$$\Delta\mathcal{L}_{\le t-1}^{(t)}\left(\Delta\mathbf{z}_t^\star(\mathcal{S}_t(I_t))\right) = \frac{1}{2}\sum_{i=1}^d m_i(\lambda_i)\,\alpha_i^2. \tag{51}$$

From equation 50 we obtain the following simple monotonicity property in the SGR strength.

**Lemma 1** (Monotonicity of SGR forgetting coefficients in $\lambda$). *Fix $i$ and treat $m_i(\lambda)$ as a function of $\lambda \ge 0$:*

$$m_i(\lambda) = \frac{\sigma_i^{(t-1)}}{(\sigma_i^{(t-1)} + 2\lambda)^2}. \tag{52}$$

*If $\sigma_i^{(t-1)} > 0$, then $m_i(\lambda)$ is strictly decreasing in $\lambda$. If $\sigma_i^{(t-1)} = 0$, then $m_i(\lambda) \equiv 0$ for all $\lambda \ge 0$.*

*Proof.* For $\sigma_i^{(t-1)} > 0$,

$$\frac{\mathrm{d}}{\mathrm{d}\lambda}\, m_i(\lambda) = \sigma_i^{(t-1)} \cdot \frac{-4}{(\sigma_i^{(t-1)} + 2\lambda)^3} < 0. \tag{53}$$

If $\sigma_i^{(t-1)} = 0$, the formula equation 50 yields $m_i(\lambda) = 0$ for all $\lambda$. $\qquad\square$

Thus, for each eigendirection with $\sigma_i^{(t-1)} > 0$, a larger SGR coefficient always reduces its contribution to the old-task loss increase. In particular, assigning the stable coefficient $\lambda_s$ to such an eigenvector yields a smaller forgetting coefficient than assigning the plastic coefficient $\lambda_p$. For directions with $\sigma_i^{(t-1)} = 0$, the forgetting coefficient is identically zero and independent of $\lambda$.

## D.3 FORGETTING CRITERIA UNDER THE SGR CONSTRAINT

We now define two natural forgetting criteria under the SGR-constrained update equation 48.

**Worst-case forgetting.** We first consider a worst-case measure over all possible new-task gradients with bounded norm in feature space. Let $\|\boldsymbol{\alpha}\|_2 \le 1$ be an upper bound on the gradient coordinates in the Hessian eigenbasis. Using equation 51, the worst-case forgetting at task $t$ under the SGR constraint is

$$F_{\mathrm{wc}}^{(t)}(I_t) := \sup_{\|\boldsymbol{\alpha}\|_2 \le 1} \Delta\mathcal{L}_{\le t-1}^{(t)}\left(\Delta\mathbf{z}_t^\star(\mathcal{S}_t(I_t))\right) = \frac{1}{2}\max_{1 \le i \le d} m_i(\lambda_i), \tag{54}$$

where $\lambda_i$ are determined by $I_t$ via equation 45.

**Average forgetting.** We also consider an average-case measure under an isotropic model for the new-task gradient direction. Suppose that $\boldsymbol{\alpha}$ is a random vector on the unit sphere or with isotropic covariance, such that $\mathbb{E}[\alpha_i] = 0$ and $\mathbb{E}[\alpha_i^2] = \frac{1}{d}$ for all $i$. Then, by equation 51,

$$\mathbb{E}\left[\Delta\mathcal{L}_{\le t-1}^{(t)}\left(\Delta\mathbf{z}_t^\star(\mathcal{S}_t(I_t))\right)\right] = \frac{1}{2}\sum_{i=1}^d m_i(\lambda_i)\,\mathbb{E}[\alpha_i^2] \tag{55}$$

$$= \frac{1}{2d}\sum_{i=1}^d m_i(\lambda_i). \tag{56}$$

This motivates the average forgetting functional

$$F_{\mathrm{avg}}^{(t)}(I_t) := \frac{1}{2d}\sum_{i=1}^d m_i(\lambda_i). \tag{57}$$

In both cases, the dependence on the subspace choice $\mathcal{S}_t$ is fully captured by the index set $I_t$ and the associated coefficients $\lambda_i \in \{\lambda_s, \lambda_p\}$.

### D.4 Optimality of the Hessian-guided stable subspace under SGR

We now show that, when the stable subspace dimension is chosen as $k = r = \mathrm{rank}(\mathbf{H}_f^{(t-1)})$, assigning the stable SGR coefficient $\lambda_s$ to all eigendirections with positive eigenvalues (i.e., choosing $\mathcal{S}_t = \mathrm{Im}(\mathbf{H}_f^{(t-1)})$) minimizes both the worst-case and average forgetting functionals $F_{\mathrm{wc}}^{(t)}(I_t)$ and $F_{\mathrm{avg}}^{(t)}(I_t)$ among all choices of $I_t$ with $|I_t| = r$ that are aligned with the eigenvectors of $\mathbf{H}_f^{(t-1)}$.

For convenience, let

$$I_{\mathrm{pos}}^{(t-1)} := \{i : \sigma_i^{(t-1)} > 0\}, \qquad I_{\mathrm{null}}^{(t-1)} := \{i : \sigma_i^{(t-1)} = 0\}, \tag{58}$$

so that $|I_{\mathrm{pos}}^{(t-1)}| = r$ and $I_{\mathrm{null}}^{(t-1)}$ indexes the zero eigenvalues.

#### Worst-case forgetting

We first analyze the worst-case forgetting functional equation 54. For each index $i$ we denote

$$a_i := m_i(\lambda_p) = \frac{\sigma_i^{(t-1)}}{(\sigma_i^{(t-1)} + 2\lambda_p)^2}, \qquad \Delta_i := a_i - m_i(\lambda_s). \tag{59}$$

By Lemma 1, if $\sigma_i^{(t-1)} > 0$ then $a_i > 0$ and $\Delta_i > 0$, whereas if $\sigma_i^{(t-1)} = 0$ then $a_i = \Delta_i = 0$. For a given index set $I_t$ with associated coefficients $\lambda_i$ as in equation 45, we can write

$$m_i(\lambda_i) = \begin{cases} a_i - \Delta_i, & i \in I_t, \\ a_i, & i \notin I_t. \end{cases} \tag{60}$$

Thus

$$F_{\mathrm{wc}}^{(t)}(I_t) = \frac{1}{2} \max_{1 \le i \le d} \left(a_i - \Delta_i \mathbf{1}\{i \in I_t\}\right). \tag{61}$$

We now specialize to the case $k = r$ corresponding to SGCL, and show that choosing $I_t = I_{\mathrm{pos}}^{(t-1)}$ is optimal.

**Theorem 1** (Worst-case optimality of the Hessian image subspace)**.** *Fix a task $t$ and let $r = \mathrm{rank}(\mathbf{H}_f^{(t-1)}) = |I_{\mathrm{pos}}^{(t-1)}|$. Consider all index sets $I_t \subset \{1, \dots, d\}$ with $|I_t| = r$, defining the SGR coefficients $\lambda_i$ via equation 45. Let $I_t^\star = I_{\mathrm{pos}}^{(t-1)}$ and*

$$\mathcal{S}_t^\star := \mathcal{S}_t(I_t^\star) = \mathrm{span}\{\mathbf{u}_i^{(t-1)} : i \in I_{\mathrm{pos}}^{(t-1)}\} = \mathrm{Im}(\mathbf{H}_f^{(t-1)}). \tag{62}$$

*Then, among all such choices of $I_t$, the worst-case forgetting functional $F_{\mathrm{wc}}^{(t)}(I_t)$ in equation 54 is minimized by $I_t^\star$, i.e.*

$$F_{\mathrm{wc}}^{(t)}(I_t^\star) \le F_{\mathrm{wc}}^{(t)}(I_t) \quad \text{for all } I_t \text{ with } |I_t| = r. \tag{63}$$

*Proof.* If $r = d$, then $\mathbf{H}_f^{(t-1)}$ is full rank and there is only one possible choice of an $r$-dimensional eigen-aligned subspace, namely the full space. In this trivial case $I_t^\star = \{1, \dots, d\}$ is the unique admissible index set and the claim holds.

We therefore assume $r < d$. Let $I_t$ be any index set with $|I_t| = r$. If $I_t = I_{\mathrm{pos}}^{(t-1)}$ there is nothing to prove, so suppose $I_t \neq I_{\mathrm{pos}}^{(t-1)}$. Then there exists at least one index $j \in I_t$ with $\sigma_j^{(t-1)} = 0$ and at least one index $\ell \notin I_t$ with $\sigma_\ell^{(t-1)} > 0$. Consider the new index set

$$\widetilde{I}_t := (I_t \setminus \{j\}) \cup \{\ell\},$$

which also satisfies $|\widetilde{I}_t| = r$.

By Lemma 1, we have $a_j = \Delta_j = 0$ and hence $m_j(\lambda_j) = 0$ for any choice of $\lambda_j \in \{\lambda_s, \lambda_p\}$. In particular,

$$m_j(\lambda_j) = m_j(\lambda_j') = 0,$$

where $\lambda_j$ and $\lambda'_j$ denote the coefficients associated with $I_t$ and $\widetilde{I}_t$, respectively. For the index $\ell$ we have $\sigma_\ell^{(t-1)} > 0$, hence $a_\ell > 0$ and $\Delta_\ell > 0$, and

$$m_\ell(\lambda_\ell) = a_\ell > a_\ell - \Delta_\ell = m_\ell(\lambda'_\ell),$$

since $\ell \notin I_t$ but $\ell \in \widetilde{I}_t$. For all other indices $i \notin \{j, \ell\}$ we have $\lambda'_i = \lambda_i$ and hence $m_i(\lambda'_i) = m_i(\lambda_i)$.

Putting these observations together, we see that the vector $\left(m_i(\lambda'_i)\right)_{i=1}^d$ is coordinatewise less than or equal to $\left(m_i(\lambda_i)\right)_{i=1}^d$, and strictly smaller in the $\ell$-th coordinate. Therefore

$$\max_i m_i(\lambda'_i) \;\leq\; \max_i m_i(\lambda_i),$$

with strict inequality whenever the maximum of the original vector is attained at index $\ell$. In particular,

$$F_{\mathrm{wc}}^{(t)}(\widetilde{I}_t) \;\leq\; F_{\mathrm{wc}}^{(t)}(I_t).$$

Starting from any index set $I_t$ with $|I_t| = r$ and repeatedly applying the above swap operation whenever $I_t \neq I_{\mathrm{pos}}^{(t-1)}$ produces a finite sequence of index sets along which $F_{\mathrm{wc}}^{(t)}$ is non-increasing and that terminates at $I_t = I_{\mathrm{pos}}^{(t-1)} = I_t^\star$. This shows that $I_t^\star$ minimizes $F_{\mathrm{wc}}^{(t)}$ over all admissible $I_t$. $\qquad\square$

Theorem 1 shows that, under the SGR constraint equation 41 and for $k = r = \mathrm{rank}(\mathbf{H}_f^{(t-1)})$, using the image of $\mathbf{H}_f^{(t-1)}$ as the stable subspace $\mathcal{S}_t$ minimizes a worst-case upper bound on the old-task loss increase over all new-task gradients with bounded norm, among all eigen-aligned stable subspaces of dimension $r$.

### AVERAGE FORGETTING

We next consider the average forgetting functional $F_{\mathrm{avg}}^{(t)}(I_t)$ defined in equation 57. Using the notation $a_i$ and $\Delta_i$ from above, we can write

$$F_{\mathrm{avg}}^{(t)}(I_t) \;=\; \frac{1}{2d} \sum_{i=1}^d m_i(\lambda_i) \;=\; \frac{1}{2d} \sum_{i=1}^d a_i \;-\; \frac{1}{2d} \sum_{i \in I_t} \Delta_i. \tag{64}$$

The first term is independent of $I_t$, so minimizing $F_{\mathrm{avg}}^{(t)}(I_t)$ over $I_t$ with a fixed cardinality $|I_t| = k$ is equivalent to maximizing the sum $\sum_{i \in I_t} \Delta_i$ over such index sets. In general this shows that the optimal index set of size $k$ is obtained by choosing the $k$ indices with largest $\Delta_i$ values.

For SGCL we again specialize to the case $k = r = |I_{\mathrm{pos}}^{(t-1)}|$, and obtain the following result.

**Theorem 2** (Average-case optimality of the Hessian image subspace)**.** *Under the same assumptions as in Theorem 1, among all index sets $I_t \subset \{1, \ldots, d\}$ with $|I_t| = r$, the average forgetting functional $F_{\mathrm{avg}}^{(t)}(I_t)$ in equation 57 is minimized by $I_t^\star = I_{\mathrm{pos}}^{(t-1)}$, i.e., by choosing $\mathcal{S}_t = \mathcal{S}_t^\star = \mathrm{Im}(\mathbf{H}_f^{(t-1)})$.*

*Proof.* If $r = d$, the claim is again trivial, since there is only one admissible choice of $I_t$. We therefore assume $r < d$.

Let $I_t$ be any index set with $|I_t| = r$. If $I_t = I_{\mathrm{pos}}^{(t-1)}$ there is nothing to prove. Otherwise there exist indices $j \in I_t$ and $\ell \notin I_t$ such that $\sigma_j^{(t-1)} = 0$ and $\sigma_\ell^{(t-1)} > 0$. Define $\widetilde{I}_t$ as in the proof of Theorem 1 by swapping $j$ and $\ell$:

$$\widetilde{I}_t \;:=\; \left(I_t \setminus \{j\}\right) \cup \{\ell\}.$$

By Lemma 1 we have $a_j = \Delta_j = 0$, hence $m_j(\lambda_j) = m_j(\lambda'_j) = 0$, and $\Delta_\ell > 0$ since $\sigma_\ell^{(t-1)} > 0$. Thus

$$\sum_{i \in \widetilde{I}_t} \Delta_i \;=\; \sum_{i \in I_t} \Delta_i - \Delta_j + \Delta_\ell \;=\; \sum_{i \in I_t} \Delta_i + \Delta_\ell \;>\; \sum_{i \in I_t} \Delta_i.$$

Consequently,

$$F_{\text{avg}}^{(t)}(\widetilde{I}_t) \;=\; \frac{1}{2d}\sum_{i=1}^{d} a_i - \frac{1}{2d}\sum_{i\in\widetilde{I}_t}\Delta_i \;<\; \frac{1}{2d}\sum_{i=1}^{d} a_i - \frac{1}{2d}\sum_{i\in I_t}\Delta_i \;=\; F_{\text{avg}}^{(t)}(I_t).$$

Starting from any $I_t$ with $|I_t| = r$ and repeatedly applying this swap whenever $I_t \neq I_{\text{pos}}^{(t-1)}$ yields a finite sequence of index sets along which $F_{\text{avg}}^{(t)}$ is strictly decreasing and that terminates at $I_t = I_{\text{pos}}^{(t-1)} = I_t^\star$. Therefore, $I_t^\star$ is the unique minimizer of $F_{\text{avg}}^{(t)}$ among all index sets $I_t$ with $|I_t| = r$. $\qquad\square$

**Summary and choice of $k$.** Under the SGR-induced local objective equation 41 and the quadratic approximation equation 39 of the old-task loss, once the feature-space Hessian $\mathbf{H}_f^{(t-1)}$ is fixed, choosing the stable subspace $\mathcal{S}_t$ as the image of $\mathbf{H}_f^{(t-1)}$,

$$\mathcal{S}_t \;=\; \text{Im}(\mathbf{H}_f^{(t-1)}) \;=\; \text{span}\{\mathbf{u}_i^{(t-1)} : \sigma_i^{(t-1)} > 0\}, \tag{65}$$

i.e., taking $k = r = \text{rank}(\mathbf{H}_f^{(t-1)})$ and marking all positive curvature directions as stable, simultaneously minimizes

- a worst-case forgetting bound $F_{\text{wc}}^{(t)}(I_t)$ over all new-task gradients with bounded norm; and

- an average forgetting functional $F_{\text{avg}}^{(t)}(I_t)$ under isotropic gradient directions,

among all $k$-dimensional linear subspaces aligned with the eigenvectors of $\mathbf{H}_f^{(t-1)}$.

In SGCL we set $k := r = \text{rank}(\mathbf{H}_f^{(t-1)})$, so that the stable subspace is exactly the image of the feature-space Hessian and the plastic subspace is its kernel. In this precise sense, the Hessian-guided stable–plastic decomposition used by SGCL is locally optimal under the SGR constraint, for both worst-case and average forgetting criteria, among all eigen-aligned decompositions with the same stable dimension.

