# OpenReview forum: "Subspace-Guided Continual Learning: Hessian Based Stable–Plastic Decomposition for Exemplar-Free Class-Incremental Learning"
_ICLR.cc/2026/Conference — ICLR 2026 Conference Desk Rejected Submission_

### Official Review · Reviewer_HLnu · 2025-10-28

**Soundness:** 2
**Presentation:** 2
**Contribution:** 3
**Rating:** 6
**Confidence:** 5

**Summary:**

This paper proposes Subspace-Guided Continual Learning (SGCL), a novel method for exemplar-free class-incremental learning (EFCIL). The key idea is to decompose the feature space into two orthogonal subspaces—stable and plastic—based on the eigenstructure of the feature-space Hessian of the cross-entropy loss. The stable subspace captures directions with high curvature that are critical for preserving previous knowledge, while the plastic subspace allows adaptation to new tasks. Building on this, the authors design two components: Subspace-Guided Regularization (SGR), which applies curvature-weighted penalties to prevent forgetting, and Subspace-Guided Prototype Alignment (SGPA), which corrects class-prototype drift and recalibrates the classifier. The method is theoretically grounded, computationally efficient, and empirically validated on several benchmarks.

**Strengths:**

1. The method provides a clear geometric interpretation of continual learning by decomposing the feature space using the eigenstructure of the feature-space Hessian. This offers a mathematically grounded alternative to heuristic or empirically tuned regularization schemes.

2. The curvature-weighted penalty mechanism (SGR) explicitly controls drift along critical directions, allowing an interpretable balance between stability and plasticity rather than a uniform penalty.

**Weaknesses:**

1. Sensitivity to hyperparameters:
The method’s performance strongly depends on the choice of the stability and plasticity weights (λₛ and λₚ). Since there’s no adaptive or self-tuning mechanism, the optimal balance must be found manually for each dataset or architecture, which limits its general applicability and robustness.

2. Approximation of the Hessian:
SGCL relies on an expected feature-space Hessian estimated through softmax probabilities. This assumes the loss landscape is locally smooth, which may not hold in highly non-linear or deeper models. As a result, the estimated curvature directions might not always represent the true “stable” directions, leading to imperfect subspace separation.

2. Dependence on classifier conditioning:
The subspace extraction process assumes that the classifier weight matrix
𝑊
𝑡
−
1
W
t−1 is full-rank and well-conditioned. In practice, as the number of classes grows, some directions can become degenerate or highly correlated, making the QR decomposition unstable and potentially distorting the identified subspace.

3. Simplified distributional assumption:
The prototype alignment module (SGPA) assumes that each class’s feature distribution is roughly Gaussian when generating synthetic samples for calibration. However, deep feature spaces are often multi-modal and non-Gaussian, which can lead to inaccurate prototype correction and weaker classifier adaptation.

4. Rigid subspace separation:
The method treats the stable and plastic subspaces as strictly orthogonal, but in real feature manifolds, these directions are often correlated. This rigid projection-based split might discard useful shared information, causing minor interference or loss of representational richness over time.

**Questions:**

1. The evaluation metrics needs more clarity. You introduced new metric, average anytime accuracy, without explaining. Normally, we Average Incremental Accuracy and Last accuracy is reported. Do they mean same?

2. Equation 10. Computing and compensating the drift between the features has already been explored in SDC and LDC. Specifically, the LDC compensates the drift using learnable projector. How this orthogonal projection is helping the model not to forget?

3. Retraining the for calibration is post-training process? What is the computational overhead for this process?


References

LDC: Gomez-Villa, Alex, et al. "Exemplar-free continual representation learning via learnable drift compensation." European Conference on Computer Vision. Cham: Springer Nature Switzerland, 2024.

SDC : Yu, Lu, et al. "Semantic drift compensation for class-incremental learning." Proceedings of the IEEE/CVF conference on computer vision and pattern recognition. 2020.

---

> ### Author Response · Authors · 2025-11-20
> **Responce to Weaknesses**
>
> We thank the reviewer for the insightful comments. Our responses are below.
>
> ---
> ### W1: Sensitivity to hyperparameters
>
> Regarding hyperparameter sensitivity, SGCL is robust to the stability weight $\lambda_s$, especially on challenging datasets. The table below shows that performance remains stable across different values of $\lambda_s$:
>
> | $\lambda_s$ | Subset (10-Task) | Subset (20-Task) | Tiny (10-Task) | Tiny (20-Task) |
> | :--- | :---: | :---: | :---: | :---: |
> | **10** | 46.58 | 33.21 | 36.78 | 30.92 |
> | **11** | 47.14 | 34.65 | 36.10 | 30.03 |
> | **12** | 48.08 | 35.11 | 35.41 | 29.24 |
>
> This is not a drawback but rather reflects precise control over stability and plasticity. Parameter tuning follows clear guidelines from our analysis (**Appendix C.3**):
>
> - **Wide range for $\lambda_s$**: The stable subspace is low-dimensional ($\le c-1$), allowing a wide tuning range for $\lambda_s$. In contrast, $\lambda_p$ operates on a larger subspace and requires finer tuning.
> - **Dataset difficulty**: For harder datasets, larger $\lambda_s$ and $\lambda_p$ values are needed to restrict feature drift and minimize forgetting.
>
> ---
>
> ### W2: Approximation of the Hessian
>
> Our feature-space Hessian (Eq. 4) is exact, not an approximation.
>
> - Theoretical Validity: We compute the Hessian w.r.t. features ($\mathbf{z}$), not parameters. For a linear classifier with Cross-Entropy loss, the loss landscape over $\mathbf{z}$ is convex. Thus, Eq. 4 provides the exact analytical curvature, independent of the backbone's depth or linearity.
> - Empirical Verification: Fig. 4(a) validates our method, showing feature drift confined to the plastic subspace. Our Hessian-based method also significantly outperforms PCA (added to Appendix C.1), confirming it better captures stability requirements.
>
> ---
>
> ### W3: Dependence on classifier conditioning
>
> We apologize for the confusion caused by the full-rank assumption in Prop. 1. This assumption was only intended to quantify the stable subspace dimension; our method itself has always relied solely on the row space of $W_{t-1}$. We have revised Prop. 1 to remove it, clarifying that the subspace dimension is simply bounded ($\le c-1$) and adapts to any degeneracy.
>
> To further address numerical stability, we empirically verified that $W_{t-1}$ remains well-conditioned. We tracked its condition number $\kappa(W_{t-1})$ on CIFAR-100, which stayed low throughout training (max $\approx 32.19$), as shown below:
>
> | Task | 0 | 1 | 2 | 3 | 4 | 5 | 6 | 7 | 8 | 9 |
> | :--- | :---: | :---: | :---: | :---: | :---: | :---: | :---: | :---: | :---: | :---: |
> | **10-Task** | 1.78 | 3.53 | 4.46 | 5.19 | 5.93 | 7.12 | 7.52 | 7.67 | 8.58 | 10.62 |
> | **20-Task (0-9)** | 1.32 | 3.28 | 4.13 | 4.60 | 4.74 | 5.61 | 5.62 | 5.52 | 6.59 | 7.04 |
> | **20-Task (10-19)** | 7.48 | 7.32 | 8.08 | 8.30 | 9.82 | 14.44 | 17.05 | 23.48 | 28.30 | 32.19 |
>
> ---
>
> ### W4: Simplified distributional assumption
>
> The Gaussian assumption is used only for classifier calibration in the SGPA module, not for feature learning. This is justified because:
>
> - Sufficiency for Calibration: Our goal is to re-balance decision boundaries. As shown in [1] (Neural Collapse), deep features cluster around class means, making Gaussian approximations of 1st/2nd-order statistics (Eq. 12) sufficient to correct classifier bias.
> - EFCIL Feasibility & Effectiveness: In the exemplar-free setting, modeling complex distributions is infeasible. Our approach is a robust surrogate. The ablation in Fig. 4c confirms this calibration significantly boosts performance, validating its effectiveness in mitigating forgetting.
>
>
> ---
>
> ### W5: Rigid subspace separation
>
> Our approach uses the intrinsic loss geometry to handle correlations and preserve the full feature manifold.
>
> - Orthogonality Decouples Correlations: The orthogonal decomposition derives from the feature-space Hessian's eigendecomposition, which naturally decouples correlations. It spans the entire feature space ($\mathcal{S} \oplus \mathcal{P} = \mathbb{R}^d$), so no information is lost.
> - Soft-Weighted Regularization: We use Subspace-Guided Regularization (SGR) for soft, curvature-aware constraints instead of a rigid freeze. Penalizing drift in $\mathcal{S}$ using Hessian eigenvalues (Eq. 7) allows necessary adaptation, preserving representational richness.
>
> [1] Papyan, et al. Prevalence of neural collapse during the terminal phase of deep learning training. In PNAS 2020.

---

> > ### Author Response · Authors · 2025-11-20
> > **Response to Questions**
> >
> > >Q1:The evaluation metrics needs more clarity. You introduced new metric, average anytime accuracy, without explaining. Normally, we Average Incremental Accuracy and Last accuracy is reported. Do they mean same?
> >
> > Yes, we have added a clarification in the paper.
> >
> > ---
> >
> > >Q2:Equation 10. Computing and compensating the drift between the features has already been explored in SDC and LDC. Specifically, the LDC compensates the drift using learnable projector. How this orthogonal projection is helping the model not to forget?
> >
> > We address this by a **unified manner**: we handle both feature drift (SGR) and prototype drift (SGPA) using the *same* stable subspace derived from Hessian analysis.
> >
> > 1.  **Unified & Efficient:** Unlike LDC which trains extra projectors, our Eq. 10 uses the **same** subspace computed for regularization. The orthogonal projection calculates a **stability score** ($S_i$) to decide if a prototype should resist drift (stable) or adapt (plastic). This reuses computation and ensures geometric consistency.
> > 2.  **Coupling & Performance:** We added experiments in Appendix C.2 comparing SGPA with LDC and SDC. This tight coupling outperforms other general methods. When pairing our SGR with different drift compensations on CIFAR-100, SGPA achieves the best results:
> >     | Method | 10 Tasks | 20 Tasks |
> >     | :--- | :---: | :---: |
> >     | SGR + SDC | 48.98% | 36.55% |
> >     | SGR + LDC | 49.10% | 35.04% |
> >     | **SGR + SGPA (Ours)** | **49.68%** | **37.23%** |
> >
> > ---
> >
> >
> > >Q3:Retraining the for calibration is post-training process? What is the computational overhead for this process?
> >
> > We apologize for the omission and have added the details to Section 4.1. The model is trained for 30 epochs using SGD with a learning rate of 1e-3, momentum 0.9, weight decay 5e-4, and a batch size of 256.
> >
> > Regarding computational overhead, the prototype re-balancing phase runs for 30 epochs after each task, fine-tuning only the classification head. Since the backbone is frozen and only a small fraction of parameters (~0.5%) are trained, the computational overhead is minimal, adding approximately 8%, 9%, and 12% to the total training time on CIFAR-100, Tiny ImageNet, and ImageNet Subset, respectively.

---

### Official Review · Reviewer_f2nm · 2025-10-30

**Soundness:** 2
**Presentation:** 3
**Contribution:** 3
**Rating:** 6
**Confidence:** 4

**Summary:**

This paper proposes a new method, SGCL, for the exemplar-free class-incremental learning (EFCIL). SGCL utilizes the Hessian martrix in the feature space to impose seperate regularization on the stable subspaces (with large loss curvature) and plastic subspaces (with smaller loss curvature) respectively via subspace projection process. The projection basis is obtained by an efficient QR factorization from the weight matrix. Subsequently, a prototype alignment process using the stability information is introduced to adrress the shift of old prototypes. Experiments are conducted to verify the performance of SGCL and its components.

**Strengths:**

1. Overall, this paper is well written and easy to follow.
2. The theorectical analysis in this paper is solid.
3. Decomposing the stable and plastic subspaces in the perspective of curvature is noval and effective.

**Weaknesses:**

1. The information of stable subspaces seems only workable for the latest task. The stable subspaces used in task $t$ relies on the Hessian matrix calculated with the data in task $t-1$, which seems that this information only works for the perservation of important knowledge of task $t-1$ and not for previous tasks. Can this regularization benefit the previous knowledge or can the subspace decomposion cover the previous tasks?
2. Limitation in proposition 1. The proposition 1 requires a full row rank $ {\mathbf{W}_{t-1}}$.  However, for learning task on a larger dataset like ImageNet-1k where $d \gt c$ , it is likely that  it will not have full row rank.  As such, proposition 1 may not hold and what will happen on the subspace decomposition? Corresponding analysis and experiments should be included. The experiment may be conducted on ImageNet-1k since the ResNet-18 has $d = 512$ and $c = 1000$.


3. The results of the plastic regularization seems not consistent. Intuitively, the regularization on plasticity can hinder learning on the new tasks. The corresponding experiment in Figure 5 also validates this, where accuracy of current tasks experiences a performance drop as the $\lambda_{p}$ increases. However, without the plastic regularization, the overall performance of SGCL drops. Why this happen and could you please include further analysis on it?
4. The methods selected in the comparative study are a bit old. Since this paper is summited to ICLR 2026, the comparative studies should contains methods proposed in 2025, for example, the DPCR [1] and etc. Also, there are several methods considering stability / plasticity feature space decoupling via gradient projection [2]. It is interesting to compare SGCL with them.

[1] Run He, et al. Semantic Shift Estimation via Dual-Projection and Classifier Reconstruction for Exemplar-Free Class-Incremental Learning. In ICML 2025.

[2] Zhen Zhao, et al. Rethinking Gradient Projection Continual Learning: Stability / Plasticity Feature Space Decoupling. In CVPR 2023.

**Questions:**

Please refer to the weaknesses.

---

> ### Author Response · Authors · 2025-11-20
>
> We thank the reviewer for the detailed and constructive feedback. Below we address each concern and summarize the modifications and new analyses we have added.
>
> ---
> ### W1: Concern on whether the stable subspace covers all past tasks
> We clarify that the stable subspace preserves knowledge from **all** previous tasks because it is derived from the cumulative classifier weights $W_{t-1}$ (which is calibrated using prototypes of all past classes Eq.12), not just the latest task. As proven in Prop. 1, the stable subspace is strictly confined within the row space of $W_{t-1}$, whose vectors define the decision boundaries for all previously learned classes.
>
> To validate this, we compare SGCL with **SGCL-Exemplar**, which computes the Hessian using a memory buffer of 50 real images/class (covering all past tasks).
>
> **Table 1: Performance on CIFAR-100**
>
> | Method | 10-Task Acc | 10-Task AAA | 20-Task Acc | 20-Task AAA |
> | :--- | :---: | :---: | :---: | :---: |
> | SGCL-Exemplar | 49.29 | 62.74 | 33.69 | 48.08 |
> | **SGCL (Ours)** | **49.68** | **62.88** | **37.23** | **49.80** |
>
> SGCL-Exemplar fails to outperform our method, confirming that the critical information is robustly encoded in the geometry of $W_{t-1}$ itself. Access to past samples provides no additional benefit over probing this geometry with current data.
>
> ---
> ###  W2: Limitation in proposition 1
>
> Thank you for pointing out this theoretical detail. We apologize for the confusion caused by the full-rank assumption in Prop. 1. This assumption was only intended to quantify the stable subspace dimension; our method itself relies solely on the row space of $W_{t-1}$. We have revised Prop. 1 in the main paper to state that the rank of the feature-space Hessian $H_f$ satisfies $r \le \min(\text{rank}(W_{t-1}), c-1)$, clarifying that the subspace dimension is simply bounded and adapts to any degeneracy.
>
> For ImageNet-1K/ResNet-18 ($d=512 < c=1000$), the stable subspace is bounded by $d$. Thus, we directly decompose the $d \times d$ Hessian $H_f$, which is more efficient ($O(d^2c)$) than the reduced Hessian method ($O(c^3)$) in this regime.
>
> Following the reviewer's suggestion, we explicitly evaluated this $d < c$ regime on ImageNet-1K (10 tasks, 100 classes per task). Under the same protocol and hyperparameters as our ImageNet-Subset experiments ($\lambda_s=20, \lambda_p=0.1$), SGCL achieves superior performance:
>
> **Table 2: Performance on ImageNet-1K(10task)**
> | Method | Acc | AAA |
> |---|---|---|
> | EFC | 42.35 |56.14|
> | LDC | 35.15 |53.88|
> | ADC | 31.34 | 50.95 |
> | **SGCL (ours)** | **44.82** | **58.49** |
>
> ---
> ### W3: The necessity of plastic regularization
> Plastic regularization is vital because the stable subspace is low-dimensional. Without a mild constraint on the plastic subspace ($\lambda_p > 0$), general feature drift in the remaining dimensions would cause forgetting. Unlike EFC[1] which adds a separate global penalty (e.g., $\|\Delta \mathbf{z}\|_2^2$), SGCL naturally integrates this control via $\lambda_p$ (details in Appendix C.3). The empirical necessity is demonstrated in the table below:
>
> **Table 3: Impact of different plasticity regularization terms on CIFAR-100 ($\lambda_s = 5$)**
> | Plasticity Term | 10-task Acc | 20-task Acc |
> |---|---|---|
> | None ($\lambda_p=0$) | 28.77 | 19.86 |
> | Global Drift ($\|\Delta \mathbf{z}\|_2^2$) | 48.73 | 35.69 |
> | **Plastic Subspace (Ours)** | **49.68** | **37.23** |
>
> ---
> ### W4: The recency of baselines (DPCR and gradient-projection feature decoupling)
> We have added the comparison with DPCR[2] in the revised manuscript. Zhao et al. [3] operate in the parameter space, heuristically constraining updates based on subspace intersections. In contrast, SGCL targets the representation space via Hessian analysis. We explicitly penalize drift in directions functionally critical for classification confidence, preserving decision boundaries via weighted penalties rather than gradient constraints.
>
> ---
> [1] Magistri,et al. Elastic Feature Consolidation For Cold Start Exemplar-Free Incremental Learning. In ICLR 2024.
>
> [2] Run He, et al. Semantic Shift Estimation via Dual-Projection and Classifier Reconstruction for Exemplar-Free Class-Incremental Learning. In ICML 2025.
>
> [3] Zhen Zhao, et al. Rethinking Gradient Projection Continual Learning: Stability / Plasticity Feature Space Decoupling. In CVPR 2023.

---

> > ### Comment · Reviewer_f2nm · 2025-11-28
> >
> > Thank you for your response! The rebuttal further strengthen my understanding of this paper and now I believe this paper is a potentially influential work with good motivation and  sound theoretical analysis. Therefore, I would like to raise my score to 8. However, at present, the system does not allow me to change my previous review and score, but once allowed, I will increase the rating. Also, ACs/SACs/PCs may refer to the updated comments. Thank you!

---

> > > ### Author Response · Authors · 2025-11-28
> > >
> > > Thank you for your follow-up and for reevaluating our paper. We sincerely appreciate your time and your support.

---

### Official Review · Reviewer_4taE · 2025-10-30

**Soundness:** 3
**Presentation:** 1
**Contribution:** 3
**Rating:** 4
**Confidence:** 3

**Summary:**

This paper focuses on the cold start class-incremental learning (CIL). It presents SGCL, an exemplar-free prototype-based approach with two synergistic components: subspace-guided regularization (SGR) and subspace-guided prototype alignment (SPGA). SGCL is motivated by the assumption that the drift of the decision plane with higher curvature need to be restricted to keep stability, and the plasticity can be improved by learning on the subspace with low-curvature decision boundaries. Therefore, SGCL divides the feature space into two subspaces (a stable subspace and a plastic subspace) based on the Hessian of the loss function. These two spaces are identified by an efficient algorithm based on the low-rank structure of the feature-space Hessian. SGR decomposes features into the stable and the plastic subspaces, while SPGPA corrects the drift of the prototypes. The authors demonstrate state-of-the-art performance on several benchmarks and provide extensive ablation studies.

**Strengths:**

1. This paper focuses on the cold-start EFCIL setting, which is a challenging reclaim in continual learning.
2. The shift from parameter to feature space and the use of Hessian curvature for a geometrically grounded stability-plasticity decomposition is an insightful contribution.

**Weaknesses:**

1. The optimum of the subspace division is neither experimentally nor theoretically verified. Although this division, based on feature space Hessian, seems to be intuitive, we still need to verify that it is optimal and better than other potential division schemes.

2. The paper lacks key information. For example,
    1. How do you select the hyper-parameters (e.g., $\lambda_s$ and $\lambda_p$) in this paper? What is the division of the training/validation/testing dataset? Is there a guideline for tuning these hyper-parameters when applying SGCL to a new dataset?
    2. What is the ImageNet-Subset dataset? How it is sampled from the full ImageNet dataset is not justified nor referenced.
    3. How we can obtain $\mathbf{W}_t$ from $\mathbf{W} _{t-1}$ is not shown in the pseudo-code (Algorithm 1). It seems that the previous classifier weight $\mathbf{W} _{t-1}$ is never used in Algorithm 1.
    4. In line 1 of Algorithm 1 (line 273), how $\theta_{\text{prev}}$ and $\mathcal{S}$ are initialized is not justified in this paper.

3. Many of the equations in this paper are confusing. For example,
    1. What is the difference between $W_t$ (line 176, 280, 232, and Figure 1) and $\mathbf{W}_t$ (line 151)?
    2. What is $\mathcal{L}$ in line 196 and beyond? Is it the same as $\mathcal{L}_{ce}$ in Eq. (1)?
    3. $\operatorname{argmin}$ is a stand-alone operator, but is separated in Eq. (1).
    4. The transpose operator is usually non-italic. However, it is italicized in this paper.
    5. Eq. (4) is not obvious. Please supplement its derivation process to make this paper easier to follow.
    6. What is the definition of $\Sigma_i$ in line 304? How it is calculated is not shown in Algorithm 1, and its cost is not considered in this paper when analyzing the computational complexity.

4. The paper confuses textual and parenthetical citations. The authors used textual citations in many places where parenthetical citations should be used.

**Questions:**

My key concerns about this paper are listed in the Weaknesses section. Besides, I list some of my concerns that do not impact my rating:
1. Can you provide visualizations of feature distributions and decision planes for different classes that change as the number of classes increases? This may help us visually observe the effect of SGCL in the feature space.
2. Can you provide the standard deviation or confidence interval of the accuracy reported in the paper, randomly varying the class order? This will enhance the statistical significance of the conclusion.
3. Using pre-trained models to solve cold-start CIL has become a popular solution in recent years [1]. Can you provide experimental results using pre-trained ViT as the backbone network?

---

[1] Zhou, Da-Wei, et al. "Continual learning with pre-trained models: A survey." *Proceedings of the Thirty-Third International Joint Conference on Artificial Intelligence*. 2024. doi:[10.24963/ijcai.2024/924](https://doi.org/10.24963/ijcai.2024/924)

---

> ### Author Response · Authors · 2025-11-20
> **Response to weaknesses**
>
> We thank the reviewer for the constructive feedback. We have addressed all concerns below, with corresponding revisions made to the manuscript.
>
> ---
> ### W1: The optimality of the subspace division.
>
> To clarify, we do not claim that our Hessian-based subspace decomposition is optimal. Rather, we choose this approach based on two considerations:
>
> (1) The Hessian matrix captures second-order information of the loss landscape;
>
> (2) Under cross-entropy loss with a linear classifier, the feature-space Hessian can be computed analytically and efficiently without computing second-order derivatives.
>
> To empirically validate the superiority of our Hessian-based approach, we have conducted an experiment comparing it with an alternative subspace division method based on Principal Component Analysis (PCA) of the within-class covariance matrix. Specifically, we extracted the top $c-1$ principal components as the stable subspace.
>
> **Table: Performance comparison of different subspace identification methods (10 tasks, Acc).**
>
> | Method | CIFAR-100 | Tiny-ImageNet |
> |--------|-----------|---------------|
> | PCA-based | 36.50 | 31.59 |
> | Hessian-based | **44.15** | **34.21** |
>
> More results and analysis have been added to the Appendix C.1.
>
> ---
> ### W2: Lacking key information.
> We thank the reviewer for pointing out these missing details, which we have now clarified in the revised manuscript.
>
> **1. Hyperparameter selection guidelines:**
>
> We select hyperparameters based on the principle $\lambda_s > \lambda_p > 0$.
>
> - **$\lambda_p$:** A small, non-zero $\lambda_p$ (typically $[0.03, 0.1]$) is essential. Since the plastic subspace is high-dimensional, leaving it unconstrained would risk excessive feature drift. By constraining $\lambda_p>0$, SGCL avoids the need for an auxiliary global L2 penalty (i.e., $\|\Delta \mathbf{z}\|_2^2$), unlike methods such as EFC [1].
>
> - **$\lambda_s$:** The stable subspace is low-dimensional yet critical, necessitating a larger $\lambda_s$ ($>1$). This parameter is less sensitive than $\lambda_p$, allowing for a broader search range.
>
> - Guidelines for New Datasets: For challenging benchmarks (e.g., ImageNet-Subset), higher values for both $\lambda_s$ and $\lambda_p$ are required to control $\Delta \mathbf{z}$. Detailed guidelines are provided in Appendix C.3.
>
> **2. About dataset:**
>
> - The ImageNet-Subset dataset is sampled from the full ImageNet dataset following the protocol established by[2]. We have added the proper citation and clarification in Section 4.1.
> - For CIFAR-100, each class contains 500 training samples and 100 test samples. For Tiny-ImageNet, each class has 500 training samples and 50 test samples. For ImageNet-Subset, each class contains 1,300 training samples and 50 test samples.
>
> **3. About Algorithm 1:**
>
> We apologize for the confusion and mistakes in Algorithm 1. We have revised the algorithm description to make it more explicit.
> - $W_t$ is *randomly initialized* and is **not** derived from $W_{t-1}$.
>
> - Both $\theta_{\text{prev}}$ and $\mathcal{S}$ refer to the parameters and stable subspace from the previous task t-1.
>
> ---
> ### W3: Notation clarifications.
>
> We appreciate the reviewer's careful reading. We address each notation issue below and have made corresponding corrections throughout the manuscript.
>
> **1. Difference between $W_t$ and $\mathbf{W}_t$:** Unified to $W_t$ (non-bold) throughout the paper.
>
> **2. Clarification of $\mathcal{L}$ in Line 196:** Refers to $\mathcal{L}_{\text{ce}}$ (Eq. 1). We have replaced all instances for consistency.
>
> **3. Formatting of argmin operator:** Corrected in Eq. (1) to standard `arg\,min`.
>
> **4. Transpose operator formatting:** Corrected to non-italic $^\top$ throughout the manuscript.
>
> **5. Derivation of Eq. (4):** Added a detailed derivation in Appendix A.
>
> **6. Definition and computation of $\Sigma_i$:**
>
> - The covariance matrix $\Sigma_i$ is defined in the text as: "$\Sigma_i$ is the covariance matrix computed from the original training features of class i." Mathematically, it is computed as:
> $$
> \mathbf{\Sigma}\_i = \frac{1}{N\_i - 1} \sum\_{j=1}^{N\_i} (\mathbf{z}\_j - \mathbf{p}\_t^i)(\mathbf{z}\_j - \mathbf{p}\_t^i)^\top
> $$
> where $N_i$ is the number of training samples for class i, and $\mathbf{p}_t^i$ is the prototype (mean feature) of class i at task t.
>
> - In SGPA, the covariance of each class prototype is computed once from the original training features and is not compensated in later tasks, leading to a per-class cost of $O(d^2 N_i)$, which has the same computational complexity as other methods such as EFC[1].
>
> ---
> ### W4: Citation formatting.
> Thank you for identifying this formatting issue. We have reviewed the manuscript and corrected all citation formats to distinguish between textual citations.
>
> [1] Magistri,et al. Elastic Feature Consolidation For Cold Start Exemplar-Free Incremental Learning. In ICLR 2024.
>
> [2] Douillard, et al. PODNet: Pooled Outputs Distillation for Small-Tasks Incremental Learning. In ECCV 2020.

---

> > ### Author Response · Authors · 2025-11-20
> > **Response to Questions**
> >
> > > Q1: Can you provide visualizations of feature distributions and decision planes?
> >
> > Thank you for this excellent suggestion. We have added Voronoi diagram visualizations in Supplemental Material showing how decision boundaries evolve in feature space as the number of classes increases.
> >
> > The visualization methodology is as follows: We compute class prototypes (mean features) from test samples and project both prototypes and individual samples into 2D space using t-SNE. Each Voronoi region represents the decision area for one class, bounded by the perpendicular bisectors between neighboring prototypes.
> >
> > ---
> >
> > > Q2: Can you provide standard deviation or confidence intervals?
> >
> > We appreciate this suggestion to enhance statistical rigor. We have conducted experiments with 5 different random seeds (varying class order) and computed the standard deviation of the final average accuracy (Acc) for all configurations. The results are summarized below:
> >
> > | Dataset | 10 Tasks | 20 Tasks |
> > |---------|----------|----------|
> > | CIFAR-100 | ±0.72 | ±1.41 |
> > | Tiny-ImageNet | ±0.79 | ±0.61 |
> > | ImageNet-Subset | ±1.32 | ±1.57 |
> >
> > These relatively small standard deviations (all below 1.6%) demonstrate that our method is robust to variations in class ordering.
> >
> > ---
> >
> > > Q3: Can you provide experimental results using a pre-trained ViT backbone?
> >
> > Thank you for this insightful suggestion. Using pre-trained Vision Transformers (ViT)[1] is indeed a common and effective solution for cold-start class-incremental learning. We have conducted additional experiments using a ViT-B/16[1] model pre-trained on ImageNet-21K[2] as the backbone, evaluated on ImageNet-A[3] with 10 tasks (uniformly divided classes).
> >
> > We compare two fine-tuning strategies: (1) **LoRA** (parameter-efficient adaptation), and (2) **Full Fine-Tuning**. For each strategy, we evaluate with and without our Subspace-Guided Regularization (SGR). The Acc results are:
> >
> > | Method | LoRA | Full Fine-Tuning |
> > |--------|------|------------------|
> > | w/o SGR | 52.93% | 17.03% |
> > | w/ SGR | 56.14% | 25.34% |
> > | **Improvement** | **+3.21%** | **+8.31%** |
> >
> > These results demonstrate that our subspace-guided regularization provides **consistent and substantial improvements** in both parameter-efficient and full fine-tuning scenarios, with particularly strong gains (+8.31%) for full fine-tuning where catastrophic forgetting is more severe. This validates the generality and effectiveness of our approach across different backbone architectures (ResNet and ViT) and training paradigms.
> >
> >
> > [1] Dosovitskiy, et al. An image is worth 16x16 words: Transformers for image recognition at scale. In ICLR 2021.
> >
> > [2] Deng, et al. A large-scale hierarchical image database. In CVPR 2009.
> >
> > [3] Hendrycks, et al. Natural adversarial examples. In CVPR 2021.

---

> ### Comment · Reviewer_4taE · 2025-11-23
> **There May Be a Fatal Flaw in the Experiments**
>
> Thank you to the authors for their timely and professional responses to my questions. **Weakness 3 and 4 are well addressed, and all my questions are clearly demonstrated.** Based on the new information, I will further clarify my rating.
>
> ### **1. Weakness 1 still exists and limits this paper from being awarded.**
> The theoretical contribution of this paper is solid and beyond the borderline. However, the inability to prove the superiority of this subspace partitioning limits it from being an awarded/oral paper.
>
> ### **2. Weakness 2 is partially addressed, but the reliability of the experiment remains a critical problem.**
> 1. The authors tuned parameters on the testing dataset without properly partitioning the validation dataset. This violates general machine learning training practices. Experimental data may lose reliability due to overfitting, especially when the hyperparameters (e.g., $\lambda_s$ and $\lambda_p$) are sensitive and highly dependent on the specific dataset.
> 2. It seems that the best hyperparameters are grid-searched according to Table 6, which is quite resource-consuming.
>
> ### **Justification of my rating**
> Although the proposed method is relatively novel, the lack of rigor in the experiments presented in this paper is fatal. I will maintain my rejection decision on this paper until it provides sufficient information to demonstrate the reliability of all its experiment results.

---

> > ### Author Response · Authors · 2025-11-25
> > **Theoretical Optimality and Experimental Rigor**
> >
> > >W1
> >
> > We thank the reviewer for the continued engagement and the high recognition of our theoretical contribution. Motivated by your insightful comment on the need to verify optimality, we have now derived a formal theoretical proof in **Appendix D: Theoretical Analysis under the SGR Constraint**, demonstrating that our Hessian-based subspace partitioning is indeed optimal under the proposed regularization framework.
> >
> > We provide a more rigorous theoretical justification for our Hessian-based subspace decomposition. Under a local quadratic approximation, the forgetting of old tasks is measured by the quadratic form $\Delta \mathbf{z}^\top \mathbf{H} \Delta \mathbf{z}$, where $\mathbf{H}$ is the feature-space Hessian. The eigendirections of $\mathbf{H}$ with positive eigenvalues are fragile, as feature drift along them causes significant forgetting, whereas directions within its kernel are safe. SGCL leverages this by defining the **stable subspace** as the image of $\mathbf{H}$ (span of eigenvectors with positive eigenvalues) and the **plastic subspace** as its kernel. As proven in Appendix D (Theorems D.1 and D.2), this decomposition is provably optimal under our SGR framework. It uniquely minimizes both worst-case and average-case forgetting compared to any other subspace partition derived from the Hessian's eigenvectors, providing a strong theoretical guarantee for its superiority over alternatives like PCA. This addresses the core of Weakness 1 by formally demonstrating why our specific choice of subspaces is not merely heuristic but principled and optimal under the stated conditions.
> >
> > ---
> > >W2
> >
> > We thank the reviewer for this critical point on experimental rigor. We sincerely apologize that our manuscript's focus on final test-set performance—a standard CL convention for fair comparison—caused confusion. We want to state clearly that **all hyperparameter search was strictly performed on a held-out validation set**, never the test set. We detail this protocol and our search process below.
> >
> > **1. Validation Set Protocol**
> >
> > Following standard practices in continual learning (e.g., [1]):
> > - Partitioning: For each class, we randomly split the training data into Training Set (90%) and Validation Set (10%).
> > - Selection: Hyperparameters were selected based on Acc on the validation set.
> > - Final Evaluation: After hyperparameter selection, we retrained the model on the full training data (Training + Validation) and reported performance on the held-out test set.
> >
> > **2. Hyperparameter Search Strategy and Results**
> >
> > Our search was systematic and theory-guided rather than exhaustive grid search. We started with CIFAR-100 as the baseline dataset, determining the optimal setting $(λ_s, λ_p) = (5, 0.03)$ through a grid search on $λ_s \in \{1, 5, 10\}$ and $λ_p \in \{0.02, 0.03\}$. For more challenging datasets (Tiny-ImageNet and ImageNet-Subset), guided by our theoretical insight that harder datasets require stronger regularization, we progressively increased the search ranges: for Tiny-ImageNet we searched $λ_s \in \{5, 10\}$ and $λ_p \in \{0.03, 0.05\}$; for ImageNet-Subset we searched $λ_s \in \{10, 20\}$ and $λ_p \in \{0.05, 0.1\}$. Importantly, we performed hyperparameter search only on the 10-task configurations, and the selected hyperparameters were directly applied to the 20-task experiments without further tuning.
> >
> > **Table 1: Hyperparameter Search on Validation Sets (10 Tasks)**
> > | Dataset | $(\lambda_s, \lambda_p)$ | Val Acc (%) |
> > |:--------|:-------------------------|:------------|
> > | **CIFAR-100** | (1, 0.02) | 36.94 |
> > | | (1, 0.03) | 37.96 |
> > | | (5, 0.02) | 43.80 |
> > | | (5, 0.03) | **44.60** |
> > | | (10, 0.02) | 43.02 |
> > | | (10, 0.03) | 43.98 |
> > | **Tiny-ImageNet** | (5, 0.03) | 34.39 |
> > | | (5, 0.05) | 32.74 |
> > | | (10, 0.03) | **35.97** |
> > | | (10, 0.05) | 31.21 |
> > | **ImageNet-Subset** | (10, 0.05) | 49.42 |
> > | | (10, 0.1) | 50.01 |
> > | | (20, 0.05) | 49.47 |
> > | | (20, 0.1) | **50.50** |
> >
> > The peaks in validation accuracy consistently align with the reported test performance.
> >
> > **3. Clarification on Table 6**
> >
> > We also clarify that Table 6 (now Table 7) was a post-hoc analysis to validate our theory (i.e., that harder datasets need stronger regularization) and was not part of the hyperparameter search. As shown above, our actual search was efficient, involving only 14 total configurations across three datasets and six experimental settings. We have clarified this in the revised manuscript.
> >
> > [1] Douillard, et al. PODNet: Pooled Outputs Distillation for Small-Tasks Incremental Learning. In ECCV 2020.

---

> ### Comment · Reviewer_4taE · 2025-11-25
> **The authors addressed all my concerns.**
>
> Thanks to the authors for addressing all my concerns. I think this paper is now ready to be a published paper of ICLR, and will increase my rating to 8.
>
> In addition, due to this paper may become influential, I suggest that the authors make their source code open-sourced after the paper is accepted.

---

> > ### Author Response · Authors · 2025-11-25
> >
> > We thank the reviewer for the very positive feedback and suggestion. We are happy to release our implementation and will make the source code publicly available upon acceptance.

---

> > ### Author Response · Authors · 2025-11-26
> >
> > We sincerely thank the reviewer for the very positive feedback and for considering to increase the rating.
> >
> > (If appropriate within the review process) we would appreciate it if the rating could be updated in the system, as this may help the AC better reflect your current assessment. In any case, we are very grateful for your supportive evaluation.

---

### Official Review · Reviewer_qRq9 · 2025-11-03

**Soundness:** 3
**Presentation:** 3
**Contribution:** 3
**Rating:** 6
**Confidence:** 4

**Summary:**

The paper aims to tackle the stability-plasticity dilemma in exemplar-free class-incremental learning (EFCIL) from a geometric perspective. The authors propose to decompose the feature space into a stable subspace and a plastic subspace by interpreting the feature-space Hessian. This allows to add extra regularization to prevent feature drifts explicitly on the stable subspace. The proposed method also incorporates a prototype-alignment module to recalibrate the classifier for old classes. The proposed method outperforms existing methods across benchmarks.

**Strengths:**

1. The paper is well-written with good motivation covering relevant works in EFCIL. The proposed method is novel and intuitive.
2. Exploiting the feature-space hessian to decompose into subspaces is interesting.
3. The extensive experiments and ablations are appreciated.

**Weaknesses:**

1. The proposed method could be further validated on fine-grained classification tasks on datasets like Cars or CUB following ADC, LDC.

2. Some analysis and comparison of the SGR regularization with full model feature distillation (without decomposing into stable and plastic directions) could highlight the significance of regularizing only the stable directions. Similarly, the impact of using the stability score $S_i$ for prototype alignment should be evaluated.

3. Since the prototype alignment stage is after the feature extractor is updated, this could be directly compared with methods like SDC, ADC and LDC which are drift correction techniques. For instance, is the SGPA approach better than these methods? How would the combination of SGR+ADC or SGR+LDC perform in comparison to SGR+SGPA?

**Questions:**

See weaknesses.

---

> ### Author Response · Authors · 2025-11-20
>
> We sincerely thank the reviewer for kind words and insightful comments.
> Below we address each weakness and summarize the new experiments we have conducted.
>
> ---
>
> ### W1: Evaluation on fine-grained datasets (CUB-200)
>
> We have added experiments on the fine-grained **CUB-200** dataset to evaluate the generality of SGCL beyond CIFAR-like benchmarks. We followed the protocol for fine-grained datasets as described in ADC.
>
> For SGCL, we adapted the backbone optimization settings (e.g., lower learning rate) to accommodate the smaller data scale, while keeping  $\lambda_s = 20$ and $\lambda_p = 0.1$ which are the same regularization weights used for ImageNet-Subset  without extensive hyperparameter tuning:
>
> **Table 1: Exemplar-free class-incremental learning on CUB-200.**
>
> | Method | 10-task Acc | 10-task AAA | 20-task Acc | 20-task AAA |
> |--------|-----------------------------|----------------------------|-----------------------------|----------------------------|
> | EFC|51.03|63.28|46.13|59.37|
> |LDC| 40.16| 54.73| 24.49| 42.67|
> | ADC| 44.65| 59.62| 19.47|39.72|
> |**SGCL (ours)**|**59.38**|**67.22**|**57.86**|**66.76**|
>
> These results have been included in Appendix C.4 of the revised paper.
>
> ---
>
> ### W2: Comparison with feature distillation and role  of the stability score $S_i$
>
> To compare our SGR regularization with full-model Feature Distillation (FD), we replaced the SGR loss with the FD loss while keeping all other settings unchanged. The results are presented below:
>
> **Table 2: Ablation of SGR regularization on CIFAR100.**
> |Regularization|10-task Acc|10-task AAA|20-task Acc| 20-task AAA|
> |-----------|-----------------------------|----------------------------|-----------------------------|----------------------------|
> |FD| 38.64| 52.77| 27.61| 40.37|
> | SGR | **49.68**|**62.88**|**37.23**|**49.80**|
>
> In SGPA, $S_i \in [0,1]$ governs the stability-plasticity balance during prototype updates. Unlike heuristic-based approaches, our method derives a principled, data-dependent $S_i$ via Hessian-based subspace decomposition. To validate its efficacy, we perform an ablation on CIFAR-100  by replacing this adaptive mechanism with fixed constants $S_i \in \{0.0, 0.25, 0.5, 0.75, 1.0\}$ while keeping the remaining architecture unchanged.
>
> **Table 3: Effect of fixing the stability score $S_i$ on CIFAR-100.**
>
> |$S_i$|10-task Acc|10-task AAA|20-task Acc|20-task AAA|
> |-----------------|---------------------|--------------------|--------------------|--------------------|
> |0.00|48.16|61.77|34.27|47.34|
> |0.25|48.44|61.81|34.58|47.63|
> |0.50|48.46|**61.87**|**34.65**|**47.71**|
> |0.75|**48.51**|61.83|34.47|47.65|
> |1.00|48.28|61.83|34.18|47.49|
>
> ---
>
> ### W3: Comparison with SDC/LDC/ADC and the Synergy of SGR+SGPA
>
> To directly address this, we conducted a controlled comparison where we kept our SGR and backbone training unchanged, modifying only the prototype update/drift compensation module: SDC (Semantic Drift Compensation) [2],  ADC (Adversarial Drift Compensation) [1], and LDC (Learnable Drift Compensation) [3]. We added further analysis in the Appendix C.2.
>
> **Table 4: CIFAR-100 with SGR but different drift-compensation modules.**
>
> |Method|10-task Acc|10-task AAA|20-task Acc|20-task AAA|
> |-----------|-----------------------------|----------------------------|-----------------------------|----------------------------|
> |SGR + SDC|48.98|62.12|36.55|48.56|
> |SGR + LDC|49.10|62.21|35.04|48.28|
> |SGR + ADC|43.00|59.05|32.76|45.08|
> |SGR +SGPA|**49.68**|**62.88**|**37.23**|**49.80**|
>
> [1]: Goswami, Dipam, et al. Resurrecting old classes with new data for exemplar-free continual learning. In CVPR 2024.
>
> [2]: Yu, Lu, et al. Semantic drift compensation for class-incremental learning.  In CVPR 2020.
>
> [3]: Gomez-Villa, Alex, et al. Exemplar-free continual representation learning via learnable drift compensation. In ECCV 2024.

---

### Author Response · Authors · 2025-11-20
**Summary of Changes**

We thank all reviewers for their constructive feedback. Below is a summary of the changes made to the manuscript:

---
### Theoretical & Methodological Improvements
1. **[R3, R4]** Revised **Proposition 1** to remove the full row rank assumption.
2. **[R2]** Refined the description of **Algorithm 1** for better clarity.
3. **[R2]** Added the explicit mathematical definition for the **class covariance matrix**.
4. **[R2]** Included the detailed derivation of **Eq. (4)** in Appendix A.

---
### New Experiments & Comparisons
5. **[R1]** Added experimental comparison with **DPCR (ICML 2025)**.
6. **[R1, R3]** Added experiments on **ImageNet-1K** and **CUB-200** datasets in Appendix C.4.
7. **[R1]** Added an analysis of **SGPA** versus other prototype drift compensation strategies in Appendix C.2.

---
### Clarifications & Experimental Details
8. **[R2, R3, R4]** Added detailed guidelines for selecting hyperparameters **$\lambda_p$ and $\lambda_s$** in Appendix C.3.
9. **[R4]** Included experimental setup details for **classifier calibration**.
10. **[R2]** Clarified the sampling protocol for the **ImageNet-Subset** dataset and added detailed training/test sample splits for all datasets.
11. **[R4]** Added further explanation for the **Acc** and **AAA** metrics.

---
### Formatting & Notation Corrections
12. **[R2]** Corrected citation formats and unified mathematical notations (e.g., $W$ vs $\mathbf{W}$, $\mathcal{L}$ vs $\mathcal{L}_{ce}$, and fixed $\operatorname{argmin}$ formatting).

13.Corrected the typo in Figure 1: "Direstion" -> "Direction".

---
### Summary of Changes in the Second Revision

**1. **[R2]** Added Theoretical Justification for Subspace Optimality (Appendix D):**
We have added **Appendix D** with **Theorem D.1 (Worst-Case)** and **Theorem D.2 (Average-Case)**, formally proving that our Hessian-based subspace decomposition is *provably optimal* for minimizing forgetting under the SGR framework.

**2. **[R2]** Clarified Experimental Protocol (Appendix C.3):**
We have revised **Appendix C.3** to: (i) provide detailed descriptions of the hyperparameter search procedure; (ii) clarify that Table 7 (formerly Table 6) was constructed post-hoc to validate our theoretical framework, not as a record of the initial search process.

---

### Note · Program_Chairs · 2026-01-17
**Submission Desk Rejected by Program Chairs**

The following references in this submission do not refer to real documents and/or have major errors in bibliographic information:

 Yuxiong Wang and Hong Zhang. Orthogonal low-rank adaptation for continual learning. In Advances in Neural Information Processing Systems (NeurIPS), volume 36, 2023.

Keke Zhu, Wang-Cheng Zhai, Yang Liu, Zheng-Jun Li, and Zheng-Jun Zha. Class-incremental learning via dual prompting. In Proceedings of the IEEE/CVF International Conference on Computer Vision, pp. 835-844, 2021c.

Marc Masana, Joost van de Weijer, Rahaf Aljundi, Cees GM Snoek, Matthias De Lange, and Tinne Tuytelaars. Class-incremental learning: A review. IEEE Transactions on Pattern Analysis and Machine Intelligence, 2024.